# Learning Concept Bottleneck Models from Mechanistic Explanations

**Antonio De Santis** [1,2*]     **Schrasing Tong**[2]     **Marco Brambilla**[1]     **Lalana Kagal**[2]

[1]Politecnico di Milano     [2]MIT CSAIL

## Abstract

Concept Bottleneck Models (CBMs) aim for ante-hoc interpretability by learning a bottleneck layer that predicts interpretable concepts before the decision. State-of-the-art approaches typically select which concepts to learn via human specification, open knowledge graphs, prompting an LLM, or using general CLIP concepts. However, concepts defined a-priori may not have sufficient predictive power for the task or even be learnable from the available data. As a result, these CBMs often significantly trail their black-box counterpart when controlling for information leakage. To address this, we introduce a novel CBM pipeline named Mechanistic CBM (M-CBM), which builds the bottleneck directly from a black-box model's own learned concepts. These concepts are extracted via Sparse Autoencoders (SAEs) and subsequently named and annotated on a selected subset of images using a Multimodal LLM. For fair comparison and leakage control, we also introduce the Number of Contributing Concepts (NCC), a decision-level sparsity metric that extends the recently proposed NEC metric. Across diverse datasets, we show that M-CBMs consistently surpass prior CBMs at matched sparsity, while improving concept predictions and providing concise explanations. Our code is available at `https://github.com/Antonio-Dee/M-CBM`.

## 1 Introduction

As AI systems become increasingly complex and embedded in high-stakes applications such as healthcare, autonomous driving, and defense, there is a growing demand for models that not only perform well but are also transparent and interpretable. To obtain explanations for AI decisions, we can generally take two approaches: (i) utilize post-hoc methods that try to gain insights into how black-box models produce their outputs, or (ii) develop inherently transparent models that can explain their decisions by design (i.e., ante-hoc explainability) (Xu et al., 2019). A promising ante-hoc approach to explainability is Concept Bottleneck Models (CBMs), which are trained to first predict an intermediate set of interpretable concepts and then use these concepts to predict the final output. Recent practice typically instantiates this concept set a-priori, either specified by human experts (Koh et al., 2020), based on knowledge graphs (Yuksekgonul et al., 2023), by prompting an LLM (Yang et al., 2023; Oikarinen et al., 2023; Srivastava et al., 2024), or using general concepts extracted from pre-trained vision-language models (Rao et al., 2024). However, concepts defined a-priori may not have sufficient predictive power for the target task or even be learnable from the available data. As a result, state-of-the-art CBMs substantially underperform their black-box counterpart when controlling for information leakage. Beyond performance, a further reason not to fix concepts a-priori is that modern ML systems often equal or exceed human expertise, creating an opportunity to use interpretability to learn from machines. For example, Schut et al. (2025) extracted concepts learned by the chess engine AlphaZero (Silver et al., 2017) and were able to teach them to grandmasters. Furthermore, mechanistic interpretability has recently made significant progress in extracting concepts learned by black-box models, in particular via Sparse Autoencoders (SAEs) (Bricken et al., 2023). Motivated by this, we ask whether CBMs built directly from a model's own learned concepts can serve as interpretable approximations of their black-box counterparts. Because these concepts originate in the backbone, we expect them to be easier to learn and to have better

---

*Work done while visiting at MIT CSAIL. Correspondence to `antonio.desantis@polimi.it`.

predictive power. To test this, we develop a novel CBM pipeline, which we refer to as Mechanistic CBM (M-CBM), and compare it to state-of-the-art CBMs in both task accuracy and its ability to learn concepts, showing significant improvements.

## 2 RELATED WORK

**Concept-based Explanations.** Early approaches for explainable AI typically rely on saliency (Selvaraju et al., 2017) or attribution maps (Sundararajan et al., 2017) that show which part of the input (e.g., regions or pixels of an image) contribute the most to a decision. By contrast, concept-based methods aim to provide explanations in terms of higher-level, human-understandable concepts (e.g., stripes for a zebra). A seminal contribution to the field was TCAV (Kim et al., 2018), a method that investigates a model's sensitivity to a user-defined concept by collecting a set of example images representing that concept. Later, De Santis et al. (2025) extended TCAV with per-instance concept attributions and saliency maps indicating where the concept is recognized. However, both methods have practical limitations as they require users to manually collect concept examples. To address this, unsupervised approaches have also been proposed (Ghorbani et al., 2019; Zhang et al., 2021; Fel et al., 2023; Bianchi et al., 2024) to automatically discover influential concepts. These methods typically perform some form of clustering of a model's activations to extract groups of semantically similar inputs or cropped patches that correspond to a concept. However, with this approach, achieving completeness (i.e., extracting a concept set sufficient to recover the model's prediction) remains a nontrivial task (Yeh et al., 2020).

**Mechanistic Interpretability.** Mechanistic interpretability (MI) aims to comprehensively *reverse-engineer* deep networks by converting their neurons and weights into interpretable features and algorithms. A central challenge to this is *polysemanticity*, i.e., neurons often respond to unrelated features, so they cannot be mapped one-to-one with concepts (Olah et al., 2020). This could allow networks to learn far more features than there are neurons, which is known as the *superposition* hypothesis (Elhage et al., 2022). Recently, Bricken et al. (2023) showed this can be addressed post-hoc by disentangling features via Sparse Autoencoders (SAEs) that learn a sparse, overcomplete dictionary of monosemantic features. Given their effectiveness in both language (Gao et al., 2025) and vision (Gorton, 2024; Thasarathan et al., 2025), we also adopt SAEs for concept extraction in our pipeline. Another emerging trend in MI is *automated interpretability*, i.e., using LLMs to automate the generation of natural language descriptions for the functional role of network components. This was first applied to explain language model neurons (Bills et al., 2023), but then also proved effective to explain vision models (Rott Shaham et al., 2024). We also use a similar approach to assign names to concepts extracted via SAEs. MI has also made progress in dissecting models into interpretable circuits (e.g., identifying algorithmic sub-structures within deep networks) via masking or patching procedures (Conmy et al., 2023). Those circuit-level analyses are currently not being used in our pipeline, but integrating them could be a promising future work.

**Concept Bottleneck Models.** Concept Bottleneck Models (CBMs) are self-explaining neural networks that learn a set of intermediate human-understandable concepts to solve a task. The term was first introduced by Koh et al. (2020), who trained CBMs using datasets featuring concept annotations. Later, Yuksekgonul et al. (2023) relaxed this requirement with post-hoc CBMs that learn a Concept Bottleneck Layer (CBL) using Concept Activation Vectors (CAVs) (Kim et al., 2018), only requiring manual selection of representative examples for each concept. Furthermore, when using a CLIP (Radford et al., 2021) backbone, they could learn concepts directly from text sourced from the ConceptNet (Speer et al., 2017) knowledge graph. Yang et al. (2023) later showed benefits in generating the concept set with LLMs. Oikarinen et al. (2023) extended this paradigm also to non-CLIP backbones using CLIP-Dissect (Oikarinen & Weng, 2023) to map concept embeddings in CLIP to any backbone. A known problem, however, that exists across all CBMs is *information leakage*, i.e., the fact that the CBL inadvertently encodes hidden class-relevant patterns beyond the concept semantics, which can be quickly learned by the final predictor to improve its accuracy (Havasi et al., 2022). This issue is quite serious, as Yan et al. (2023) even showed that replacing concepts with random words can achieve similar accuracy. Information leakage also results in unsatisfying explanations, in which the most important concepts contribute significantly less than the sum of all other concepts, making the model basically a black-box. Furthermore, while for classical CBMs, leakage can at least be quantified using metrics based on ground-truth concept labels (Havasi et al., 2022;

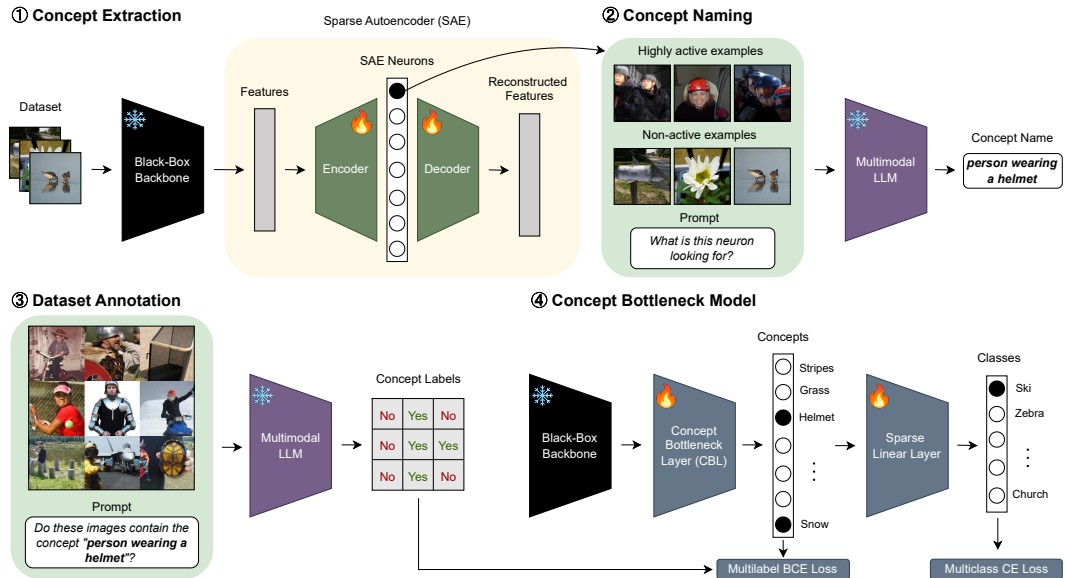

Figure 1: Overview of the M-CBM pipeline. (1) Given a trained black-box backbone, we extract its features and learn sparse, disentangled concept directions using a Sparse Autoencoder (SAE). (2) A Multimodal LLM is prompted with examples of highly activating and non-activating images to assign concept names to each SAE neuron. (3) The MLLM then annotates a subset of the dataset containing an equal split of active and non-active examples, indicating the presence or absence of each concept in selected images. (4) Using these concept annotations, we train a Concept Bottleneck Layer (CBL) and a sparse linear classifier to predict target classes from the learned concepts.

Zarlenga et al., 2023), this is not trivial for CBMs that learn concepts automatically. To address this, Srivastava et al. (2024) introduced the Number of Effective Concepts (NEC) as a metric to measure and control how many concepts CBMs use to make a prediction, effectively reducing information leakage. In this work, we also follow this idea and use the Number of Contributing Concepts (NCC), a generalization of NEC, to control for leakage and explanation conciseness. More details on NEC and NCC are provided in Section 4. Srivastava et al. (2024) also introduced VLG-CBM, a CBM pipeline that uses GroundingDINO (Liu et al., 2025), an open-vocabulary object detector, to automatically annotate a dataset with LLM-generated concepts. The CBL is then trained on these annotations in a multilabel setting. However, leakage still arises from the annotation being class-conditioned, as we show in Section J. Another limitation of these CBM paradigms is that LLM-generated concept sets may not have sufficient predictive power for the target task or not even be learnable from the available data, requiring the inclusion of uninterpretable components in the bottleneck to recover accuracy (Yuksekgonul et al., 2023; Zhang et al., 2025; Shang et al., 2024). Sometimes concepts can also be non-visual (Roth et al., 2023) (e.g., "spicy", "loud"), making explanations less transparent. Instead, we propose extracting and using the black-box model's own learned concepts, rather than guessing with an LLM. A first step in this direction is DN-CBM (Rao et al., 2024), which learns concepts from CLIP with an SAE and uses its hidden layer as CBL, naming the concepts by selecting the nearest text embeddings to the decoder vector. However, this paradigm can only be applied with a CLIP backbone, limiting its accuracy across datasets, as we show in Section 6, and CLIP dependence can still introduce non-visual concepts.

## 3 METHODOLOGY

In this section, we introduce our methodology for transforming any black-box model into an interpretable-by-design CBM. Our approach, which we refer to as Mechanistic CBM (M-CBM), extracts human-interpretable concepts from a trained black-box model, assigns names and annotations using a Multimodal Large Language Model (MLLM), and then trains a sequential CBM (Koh et al., 2020) using these concepts. An overview of the whole pipeline is provided in Figure 1.

**Concept Extraction.** Given a black-box backbone $\phi$ trained on an arbitrary dataset $\mathbb{D}$, the first step of our methodology is to decompose the features learned by the model during training into a set of interpretable concepts. To achieve this, we use the Sparse Autoencoder (SAE) approach, which was recently popularized in the mechanistic interpretability literature (Bereska & Gavves, 2024) and has proven effective to disentangle model features into interpretable concepts for both vision (Gorton, 2024; Thasarathan et al., 2025) and language models (Bricken et al., 2023; Huben et al., 2024).

An SAE is a neural network trained to reconstruct its input features while enforcing sparsity in the hidden representation (see step ① in Figure 1). In our case, the input features are the activations $\boldsymbol{a}^{(i)} = \phi(\boldsymbol{x}^{(i)}) \in \mathbb{R}^n$ of the backbone $\phi$ for each sample $\boldsymbol{x}^{(i)}$ in the training set $\mathbb{D}$. Following Bricken et al. (2023), the SAE subtracts an input bias $\boldsymbol{b}_D$ and then passes the resulting vector to an encoder with weights $\boldsymbol{W}_E$, bias $\boldsymbol{b}_E$, and ReLU activation, obtaining the hidden layer $\boldsymbol{h} \in \mathbb{R}^m$:

$$\boldsymbol{h} = \text{ReLU}\left(\boldsymbol{W}_E^\top (\boldsymbol{a} - \boldsymbol{b}_D) + \boldsymbol{b}_E\right)$$

In the sparse hidden layer $\boldsymbol{h}$, ideally, each neuron learns to recognize a distinct concept. A decoder with weights $\boldsymbol{W}_D$ and bias $\boldsymbol{b}_D$ then maps $\boldsymbol{h}$ back to the reconstructed features $\hat{\boldsymbol{a}}$:

$$\hat{\boldsymbol{a}} = \boldsymbol{W}_D^\top \boldsymbol{h} + \boldsymbol{b}_D$$

where $\boldsymbol{W}_E \in \mathbb{R}^{n \times m}$, $\boldsymbol{W}_D \in \mathbb{R}^{m \times n}$, and typically for large datasets $m \gg n$ to account for the superposition hypothesis (Elhage et al., 2022), i.e., the fact that neural networks tend to learn more concepts than the neurons they have. The input and output biases $\boldsymbol{b}_D$ are opposite in sign and equal in magnitude. While Bricken et al. (2023) train SAEs with expansion factors (defined as $m/n$) ranging from $1x$ to $256x$, we avoid going above $4x$ to keep the annotation step computationally feasible. To train the SAE, we minimize the following objective:

$$\mathcal{L}_{\text{SAE}} = \frac{1}{|\mathbb{D}|} \sum_{i=1}^{|\mathbb{D}|} \|\boldsymbol{a}^{(i)} - \hat{\boldsymbol{a}}^{(i)}\|_2^2 + \lambda_{\text{SAE}} \|\boldsymbol{h}^{(i)}\|_1 \tag{1}$$

where $\lambda_{\text{SAE}} > 0$ is a hyperparameter that controls the strength of the sparsity penalty on the hidden representation. We also monitor the average $\ell_0$ norm (i.e., the number of non-zero activations) to ensure $\ell_0 \ll n$, as recommended by Bricken et al. (2023).

SAE training often leaves many neurons in the hidden layer $\boldsymbol{h}$ dead (never activated for any training sample) or nearly dead (activate only very rarely). To ensure that our set of candidate concepts is both meaningful and computationally efficient for subsequent annotation, we perform a filtering step to remove such neurons. To define the threshold for identifying nearly dead neurons, we measure, for each unit in $\boldsymbol{h}$, the number of training samples for which it is active. We then select a cutoff value such that removing all units below this threshold does not reduce the recovered cross-entropy loss of the black-box model, defined as $1 - \frac{\mathcal{L}_{\text{BB}}(\hat{\boldsymbol{a}}) - \mathcal{L}_{\text{BB}}(\boldsymbol{a})}{\mathcal{L}_{\text{BB}}(\boldsymbol{0}) - \mathcal{L}_{\text{BB}}(\boldsymbol{a})}$, by more than a tolerance of $\sim 1\%$. This procedure ensures that only neurons with negligible contribution to predictive performance are pruned. This metric was also used to evaluate SAE quality in prior work (Bricken et al., 2023; Rajamanoharan et al., 2024; Gao et al., 2025). More details on SAEs in Appendix B.

**Concept Naming.** After pruning, each remaining neuron in the SAE hidden layer $\boldsymbol{h}$ is treated as a candidate concept, where we denote by $h_j$ the $j$-th hidden SAE neuron. To assign human-interpretable names, we adopt an automated procedure inspired by recent work on mechanistic interpretability of language model neurons (Bills et al., 2023). For each candidate concept, we first select a set of inputs $\boldsymbol{x} \in \mathbb{D}$ that maximally activate the corresponding neuron $h_j$. For these inputs, we also highlight the spatial regions that contribute the most to the activation, similarly to Rott Shaham et al. (2024). To compute these concept saliency maps, we use the method introduced by De Santis et al. (2025) (i.e., weighted average of feature maps using $\boldsymbol{W}_D$ as concept weights and followed by ReLU). To provide a contrastive signal, we additionally sample a set of non-activating examples, of which half are drawn at random from $\mathbb{D}$, and half are selected as the most cosine similar to the activating examples to enhance discrimination of fine-grained visual features. The paired examples are provided to an MLLM, GPT-4.1 in our experiments, which is prompted to produce a concise natural-language description of the concept that the neuron represented by $h_j$ is responding to. At this stage, we also explicitly instruct the model not to use class names as concepts and re-try if it does not adhere. Step ② of Figure 1 shows an example of what the MLLM receives as input. In our experiments, we used 10 activating examples and 10 non-activating ones.

Finally, since we do not want duplicate or semantically equivalent concepts, we perform a merging step similar to Oikarinen et al. (2023), in which we embed all proposed textual names using a pre-trained embedding model and merge those with very high cosine similarity (i.e., above $0.98$). We use OpenAI's *text-embedding-3-large* in our experiments. To make the embeddings context-aware, we also wrap each concept name in the following template before inserting it into the embedding model: "This is a visual concept in the context of {*domain*}: {*concept*}". The variable {*concept*} contains the concept name, while {*domain*} specifies the dataset domain (e.g., bird species, skin lesions). For simplicity, Figure 1 omits the merging step.

**Dataset Annotation.**   With concept names assigned, we proceed to build a partially annotated dataset to train the Concept Bottleneck Layer (CBL). This step is necessary because a concept name is only a hypothesis, rather than a faithful description of the corresponding SAE neuron's functional role. Since such hypotheses are often difficult to validate (Sharkey et al., 2025), we do not consider it ideal to use the SAE hidden layer directly as a bottleneck when interpretability is the primary goal.

Let $\mathbb{C} = \{c_1, \ldots, c_K\}$ denote the final set of concepts. For each concept $c_k$, the goal is to obtain binary presence/absence labels on a subset of images $\boldsymbol{x} \in \mathbb{D}$. Since exhaustive annotation of the full dataset is not computationally feasible at the time of writing this paper, we annotate up to $1000$ samples per concept. The annotation procedure is performed by prompting the MLLM with batches of $25$ images arranged in a $5 \times 5$ grid, which is much cheaper than one image at a time, but also slightly less accurate (see Appendix G). The model is asked to indicate, for each of the $25$ grid images, whether the concept is present or absent. See step ③ of Figure 1 for a high-level overview of the annotation procedure. Each call also includes a grid of the top-25 most activating images for the corresponding SAE neuron, which serve as a reference together with the textual concept name. To select the subset of images for annotation, we first select up to $500$ active samples per concept. The active set is defined as all inputs for which $h_j > 0$. From this set, we select samples whose activation lies above the 95th percentile of the set. If fewer than $500$ samples exceed this percentile, we take the top-500 activations overall within the active set. If the neuron has fewer than $500$ active samples in total, we take all available examples, rounding the number down to the nearest multiple of $25$ to match the batch annotation protocol. For merged neurons, activations are normalized across the group and treated as a single unit when computing percentiles. We then select an equal number of non-active samples, of which half are drawn uniformly at random and half are chosen as the most cosine similar to the active samples, similarly to the naming procedure. Furthermore, to avoid biasing concepts toward particular classes, both active and non-active sets are stratified across class labels. Each batch of 25 images also contains a balanced mix of active and non-active examples. At the end of this annotation step, we obtain a set of around 1000 annotated samples for each concept, containing both presence and absence cases across both training and test data. An image may be annotated for more than one concept or for none. Formally, for each image $\boldsymbol{x}^{(i)} \in \mathbb{D}$, the annotation procedure creates a ternary vector of concept labels $\boldsymbol{z}^{(i)} \in \{-1, 0, 1\}^K$ with the following entries:

$$z_k^{(i)} = \begin{cases} 1 & \text{if } c_k \text{ is annotated as } \textit{present} \text{ in } \boldsymbol{x}^{(i)} \\ 0 & \text{if } c_k \text{ is annotated as } \textit{absent} \text{ in } \boldsymbol{x}^{(i)} \\ -1 & \text{if } c_k \text{ is } \textit{not annotated} \text{ for } \boldsymbol{x}^{(i)} \end{cases} \tag{2}$$

**Concept Bottleneck Model.**   After generating the concept labels, we proceed with training a sequential CBM (Koh et al., 2020). As shown in step ④ of Figure 1, the CBM has three components: (i) a frozen backbone $\phi$ that maps an input image to a feature vector, (ii) a Concept Bottleneck Layer (CBL) $g$ that predicts the presence of $K$ named concepts from those features in a multi-label setting, and (iii) a sparse linear classifier $f$ that predicts the class from the concept outputs.

For each input $\boldsymbol{x}^{(i)}$ the frozen backbone produces $n$-dimensional features $\boldsymbol{a}^{(i)} = \phi(\boldsymbol{x}^{(i)}) \in \mathbb{R}^n$. The CBL $g : \mathbb{R}^n \to \mathbb{R}^K$ takes these features as input and outputs concept logits, then a sigmoid produces probabilities $\hat{\boldsymbol{z}}^{(i)} = \sigma(g(\boldsymbol{a}^{(i)})) \in [0,1]^K$. From the annotation pipeline, each image carries a ternary concept vector $\boldsymbol{z}^{(i)} \in \{-1, 0, 1\}^K$ indicating present (1), absent (0), or not annotated ($-1$). Since not every image-concept pair is labeled, we train $g$ only on the entries we know. Let $\Omega = \{(i,k) : z_k^{(i)} \in \{0,1\}\}$ be the set of annotated pairs. The CBL is optimized to minimize a masked Binary Cross-Entropy (BCE) loss that averages over $\Omega$:

$$\mathcal{L}_{\text{CBL}} = \frac{1}{|\Omega|} \sum_{(i,k) \in \Omega} \text{BCE}\left(\hat{z}_k^{(i)}, z_k^{(i)}\right) \tag{3}$$

Entries with $z_k^{(i)} = -1$ are effectively ignored in the loss computation. Therefore, images without any concept annotation (all entries $-1$) are not used to train the CBL. Furthermore, since positives are often rarer than negatives, we weight each concept in the BCE by the ratio of its class imbalance.

To map concepts to classes, we follow prior work (Srivastava et al., 2024; Yuksekgonul et al., 2023; Oikarinen et al., 2023) and train a sparse linear classifier on concept logits (i.e., CBL's pre-sigmoid outputs), optimized using the GLM–SAGA solver (Wong et al., 2021). Since GLM–SAGA assumes standardized input features, we $z$-normalize (zero mean and unit variance) the concept logits and use these to predict the classes. With $g$ frozen, we define a fully connected layer $f : \mathbb{R}^K \to \mathbb{R}^C$ with weights $\boldsymbol{W}_F \in \mathbb{R}^{K \times C}$ and bias $\boldsymbol{b}_F \in \mathbb{R}^C$, where $C$ is the number of output classes, and minimize the following Cross-Entropy (CE) loss with an elastic-net (Zou & Hastie, 2005) penalty:

$$\mathcal{L}_{\text{CLF}} = \frac{1}{|\mathbb{D}|} \sum_{i=1}^{|\mathbb{D}|} \text{CE}\left(f \circ g \circ \phi(\boldsymbol{x}^{(i)}), \boldsymbol{y}^{(i)}\right) + \lambda_{\text{CLF}} R_\alpha \tag{4}$$

where $\boldsymbol{y}^{(i)}$ represents the one-hot ground-truth class label for sample $\boldsymbol{x}^{(i)}$ and $R_\alpha = (1 - \alpha) \frac{1}{2} \|\boldsymbol{W}_F\|_2^2 + \alpha \|\boldsymbol{W}_F\|_1$ denotes the elastic-net penalty. Following Wong et al. (2021), we use $\alpha = 0.99$, while $\lambda_{\text{CLF}}$ is tuned to obtain a target sparsity.

## 4 NUMBER OF CONTRIBUTING CONCEPTS (NCC)

Prior work has shown that sparse layers are more interpretable (Wong et al., 2021; Yuksekgonul et al., 2023; Oikarinen et al., 2023), and Srivastava et al. (2024) also showed that sparsity is inversely correlated with information leakage. They demonstrated that a dense linear classifier built on top of a random (i.e., untrained) CBL can recover black-box accuracy if the number of concepts $K$ approaches or exceeds the backbone feature dimension $n$, but this effect decreases with higher sparsity. Related studies (Shang et al., 2024; Yan et al., 2023) similarly report that when the concept set is large enough (e.g., $K \gtrsim n/2$), dense linear classifiers can preserve black-box accuracy by re-estimating the backbone activations, therefore even using random words as concepts can match the accuracy obtained with concepts defined by LLMs or humans.

While high sparsity improves interpretability and limits leakage, it naturally tends to correlate with lower accuracy (Wong et al., 2021; Oikarinen et al., 2023; Srivastava et al., 2024), making CBM comparison incomplete if only accuracy is reported. To address this, Srivastava et al. (2024) introduced an evaluation metric named NEC, which is defined as the average number (per-class) of non-zeros in the weights $\boldsymbol{W}_F$ of the final layer $f$. They train $f$ at different regularization strengths $\lambda_{\text{CLF}}$ and accuracies are compared at equal NEC. This is convenient for enabling a fair comparison between CBMs, but it also has limitations. Controlling NEC forces concise decision explanations, but it does so by linearly restricting the effective concept vocabulary as the number of classes decreases. For instance, with three classes, NEC=5 forces $K \leq 15$ (or even $K = 5$ for binary classification with single output head) after training so that on average predictions are explained by $\sim 5$ concepts. However, in datasets with substantial intra-class diversity (e.g., peeled or in-field pineapples are the same class in ImageNet), a class may require a rich concept vocabulary (i.e., larger $K$) to cover its different contexts, even though only a subset of them is needed to predict an individual image.

With this in mind, we introduce a generalization of NEC, named Number of Contributing Concepts (NCC), which does not impose a hard cap on $K$ but still enforces concise explanations by measuring sparsity at decision-level using concept contributions rather than weights count. To measure the contribution of concept $k$, for class $r$ and image $i$, we must consider the magnitude of both the concept logit $g(\boldsymbol{a}^{(i)})]_k$ and its weight $[\boldsymbol{W}_F]_{k,r}$ towards class $r$. We then define the absolute contribution of a concept to a class as $u_{k,r}^{(i)} = \left| [g(\boldsymbol{a}^{(i)})]_k \cdot [\boldsymbol{W}_F]_{k,r} \right|$. Ideally, we want the model to recognize a class with only a small subset of concepts that cover the vast majority of the total absolute contribution, or, in other words, explain the vast majority of the decision. Let $u_{(s),r}^{(i)}$ denote the $s$-th largest absolute contributing concept, and fix a coverage level $\tau \in [0, 1]$. We define NCC as:

$$\text{NCC}_\tau = \frac{1}{|\mathbb{D}| \, C} \sum_{i=1}^{|\mathbb{D}|} \sum_{r=1}^{C} \min\left\{ \kappa \in \{0, \dots, K\} : \sum_{s=1}^{\kappa} u_{(s),r}^{(i)} \geq \tau \sum_{k=1}^{K} u_{k,r}^{(i)} \right\}$$

While here we average NCC over all $C$ classes, it can also be alternatively computed considering only the model's predicted class. Intuitively, $\text{NCC}_\tau$ is the average number of concepts required to

explain at least a $\tau$ fraction of the prediction of a class. For example, an NCC=5 with $\tau = 0.95$, means that, on average, just 5 concepts explain $\geq 95\%$ of the decision. For controlling NCC, we fix a $\tau$ and follow the approach of Srivastava et al. (2024), training $f$ at different $\lambda_{\mathrm{CLF}}$ and compare CBM accuracies at equal NCC levels. In practice, targeting a lower NCC generally means trading accuracy for explanation conciseness and vice versa.

## 5 EXPERIMENTAL SETUP

**Baselines.** We compare M-CBM with three state-of-the-art CBMs: LF-CBM (Oikarinen et al., 2023), VLG-CBM (Srivastava et al., 2024), and DN-CBM (Rao et al., 2024). For VLG-CBM, we compare with a class-agnostic annotation variant, which we refer to as VLG-CBM$_{\mathrm{CA}}$, rather than the original pipeline. In the original VLG-CBM, concepts are assigned to classes *before* annotation and are annotated only on images of their assigned classes. While this design reduces annotation cost, coupling concepts to classes can introduce substantial information leakage. We verify this on CUB in Figure 2. Using random words as concepts, VLG-CBM reaches black-box level accuracy already at around NCC=1.5, showing that in this setup, performance is insensitive to both sparsity and concept semantics. Intuitively, this happens because learning a concept that is labeled as positive only on images of a class is nearly equivalent to learning that class directly. When we remove class conditioning by annotating each concept across all images (VLG-CBM$_{\mathrm{CA}}$), accuracy drops substantially for both random and real concepts, and the expected interpretability–accuracy trade-off reappears. We performed the same experiment with our M-CBM, and the performance of substituting concepts with random words is similar to using random words in VLG-CBM$_{\mathrm{CA}}$. However, when real concepts are used, our M-CBM outperforms VLG-CBM$_{\mathrm{CA}}$ at high sparsity (NCC=3 to 5), while for low sparsity (NCC=10+), accuracy becomes similar to the random words variant due to information leakage. Further implementation details for this experiment are provided in Appendix C.

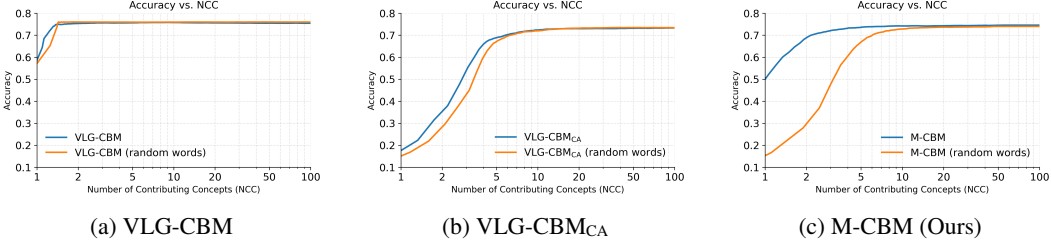

| (a) VLG-CBM | (b) VLG-CBM$_{\mathrm{CA}}$ | (c) M-CBM (Ours) |

Figure 2: Accuracy vs NCC ($\tau = 0.95$) on CUB. (a) With class-conditioned annotation, VLG-CBM reaches near black-box accuracy with NCC=1.5 (i.e., using only 1 to 2 concepts per prediction). The same happens using random concept names, showing evidence of leakage. (b) Making annotation class-agnostic (VLG-CBM$_{\mathrm{CA}}$) restores the accuracy–interpretability trade-off, with real concepts slightly beating random words at low NCC. (c) M-CBM outperforms both VLG-CBM$_{\mathrm{CA}}$ and the random baselines at low NCC, while leakage is significant for both methods as NCC increases.

**Setup.** We evaluate on three standard image classification datasets that vary in domain and class count: CUB (Wah et al., 2011), ISIC2018 (Codella et al., 2019; Tschandl et al., 2018), and ImageNet (Deng et al., 2009). CUB contains $\sim 6k$ training images and $\sim 5.8k$ test images of 200 fine-grained bird species. As backbone for this dataset, we use the pre-trained ResNet18 from *pytorchcv*. ISIC2018 contains dermatoscopic images of pigmented lesions categorized in 7 classes, split into $\sim 10k$ train, 193 validation, and $\sim 1.5k$ test. Given a high class imbalance, we report both accuracy and balanced accuracy for this dataset. Given the lack of public pre-trained models, we train a ResNet50 (weighting each class by its imbalance ratio) and use it as a backbone. ImageNet includes $1k$ classes with $\sim 1.3M$ training and $50k$ test images for general image classification. As backbone, we use the pre-trained ResNet50 from *torchvision*. Furthermore, for ImageNet and CUB, we extract $10\%$ from the train and use it as a validation set. Regarding DN-CBM, since it only supports a CLIP backbone, we evaluate it using both the ResNet50 and ViT-B/16 backbones. As discussed in Section 4, we compare under the same NCC. We use $\tau = 0.95$ and measure accuracies at NCC=5 and NCC=avg, with the latter being the average of the levels: 5, 10, 15, 20, 25, 30.

**Compute Resources.**  We trained all neural components (SAE, CBL, and GLM-SAGA) on an HPC cluster using an NVIDIA H200 on a multi-core node (32 cores and 512GB of RAM). On CUB and ISIC2018, each stage takes 5-20 minutes, while 5-10 hours for ImageNet. The dominant step in terms of cost and runtime is the annotation with GPT-4.1 API, which takes around 2 minutes and costs USD 0.14 per concept. Concept naming was lighter, taking around 10-20 seconds and USD 0.02 per concept, while concept merging costs were negligible. These costs scale linearly with the concept number, which was 278, 73, and 2648, respectively for CUB, ISIC2018, and ImageNet.

## 6  RESULTS AND DISCUSSION

**Accuracy Comparison.**  We report results in Table 1. Our M-CBM consistently achieves the highest accuracy across datasets and NCC values. An expected interpretability-accuracy trade-off is also visible across all methods, as accuracy always increases when NCC is higher (i.e., explanations are less concise). DN-CBM consistently performs poorly, especially at NCC=5, indicating that a small subset of generic CLIP concepts may be insufficient to predict a class across datasets. VLG-CBM$_{CA}$ shows better accuracy than LF-CBM and DN-CBM, but annotating per-concept the entire dataset with GroundingDINO makes it computationally prohibitive at ImageNet scale ($\sim 300$ GPU-days). In contrast, M-CBM uses SAE activations to pre-select candidate images per concept, so that we only need to annotate $\sim 1k$ images per concept. We exclude class-conditioned VLG-CBM from the comparison because, due to leakage, it is effectively a black-box (see Section J).

Table 1: Accuracy comparison at NCC=5 and NCC=avg with best model in bold. The results for M-CBM are averaged over 3 seeds with same annotations. N/A denotes computationally unfeasible.

| Dataset | CUB | | ISIC2018 | | | | ImageNet | |
|---|---|---|---|---|---|---|---|---|
| Metrics | Accuracy | | Accuracy | | Balanced Accuracy | | Accuracy | |
| Black-box | 76.67% | | 79.37% | | 75.37% | | 76.15% | |
| Sparsity | NCC=5 | NCC=avg | NCC=5 | NCC=avg | NCC=5 | NCC=avg | NCC=5 | NCC=avg |
| LF-CBM | 58.08% | 71.09% | 61.44% | 67.55% | 64.29% | 67.30% | 62.20% | 69.08% |
| DN-CBM$_{RN}$ | 38.21% | 48.98% | 35.38% | 54.61% | 39.85% | 52.85% | 46.71% | 57.24% |
| DN-CBM$_{ViT}$ | 48.12% | 66.19% | 43.92% | 56.08% | 42.47% | 53.56% | 60.23% | 69.98% |
| VLG-CBM$_{CA}$ | 69.12% | 72.25% | 64.55% | 72.61% | 64.63% | 70.80% | N/A | N/A |
| **M-CBM** | **73.70%** | **74.18%** | **72.75%** | **75.51%** | **70.14%** | **71.54%** | **72.18%** | **73.64%** |
| **(Ours)** | $\pm 0.13\%$ | $\pm 0.06\%$ | $\pm 0.10\%$ | $\pm 0.08\%$ | $\pm 0.09\%$ | $\pm 0.05\%$ | $\pm 0.21\%$ | $\pm 0.15\%$ |

**Evaluating Concept Prediction.**  We assess how well each method can learn its own concepts by also annotating the test set. Because these labels are not ground truth, high scores do not guarantee that the model is learning the concepts as intended, but only that they are at least internally consistent and learnable. Especially for ISIC2018, we found that LLM-generated concept sets are often non-visual (e.g., "warm to the touch") or not in the data (e.g., "medical report"). Since M-CBM uses concepts extracted from the backbone, we expect some benefits in the concept predictions, which is what we see in Table 2. Another factor that could contribute to the lower performance is the capability of GroundingDINO to annotate correctly, which may be inferior to asking GPT-4.1, especially for medical images. However, due to a lack of ground truth, this remains challenging to quantify.

Table 2: ROC-AUC evaluation of concept predictions on test set. Each method is evaluated on its own concepts. We report the macro-average across concepts and the average of the worst 10%.

| Dataset | CUB | | ISIC2018 | | ImageNet | |
|---|---|---|---|---|---|---|
| Metrics | ROC-AUC | | ROC-AUC | | ROC-AUC | |
| | Macro | Worst-10% | Macro | Worst-10% | Macro | Worst-10% |
| VLG-CBM$_{CA}$ | 62.03% | 45.60% | 73.37% | 52.92% | N/A | N/A |
| **M-CBM (Ours)** | **90.04%** | **79.05%** | **80.57%** | **66.98%** | **88.90%** | **78.36%** |

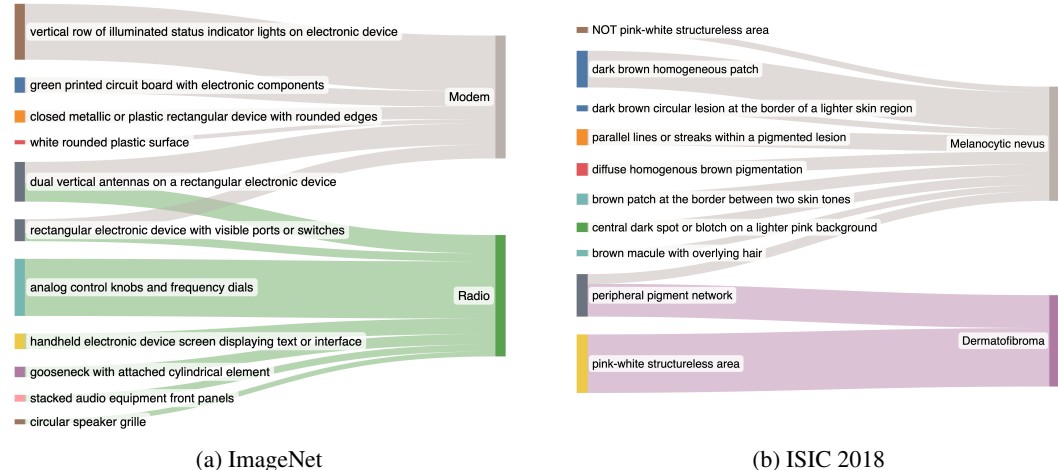

(a) ImageNet        (b) ISIC 2018

Figure 3: Sankey plots of concept–class weights of our M-CBM at NCC=5. Concepts on the left and classes on the right. Concepts with negative weights are labeled as "NOT concept".

**Explanations.** We illustrate the behavior of M-CBMs through global (class-level) and local (instance-level) explanations, using models at NCC=5. Using the final layer weights $\boldsymbol{W}_F$, we can visualize how concepts globally contribute to classes. In Figure 3, we show these weights using Sankey diagrams, with "NOT concept" indicating a negative weight. For clarity, we include only concepts with $|W_F| > 0.1$. On ImageNet, the model's behavior aligns with intuition. The classes "Modem" and "Radio" share concepts related to ports/switches and antennas, while they are mainly differentiated by the presence of indicator lights for class "Modem" versus control knobs for class "Radio". On ISIC2018, the model learns a richer concept set for "Melanocytic nevus" than for "Dermatofibroma", which could be explained by the large class imbalance. Still, the few concepts learned for "Dermatofibroma" seem reasonable considering dermatological literature (Zaballos et al., 2006). Some minor concepts for "Melanocytic nevus", such as skin-tone–related terms, are less clear. This likely arises from the concept naming (step ②), where visually highlighting the concept (in this case, the skin around the nevus) in the image can introduce mild artifacts that GPT-4.1 over-interpreted. CBMs can also explain individual predictions by showing, for an input $x^{(i)}$, the contribution of concepts to a class $r$. This contribution is computed directly by multiplying the logit of the $k$-th concept $g(\boldsymbol{a}^{(i)})]_k$ with its corresponding weight $[\boldsymbol{W}_F]_{k,r}$ towards class $r$. Concepts with a negative logit are indicated as "NOT concept". We show two examples in Figure 4, including a correct CUB prediction and a misclassification on ISIC, where the model incorrectly sees "clustered blue-gray ovoid nests", leading to a "Basal Cell Carcinoma" prediction. Zeroing this concept flips the decision to the correct class. In both cases, we see that the decision is largely explained by the top 4-5 concepts. More examples of explanations are provided in Appendix F and J.

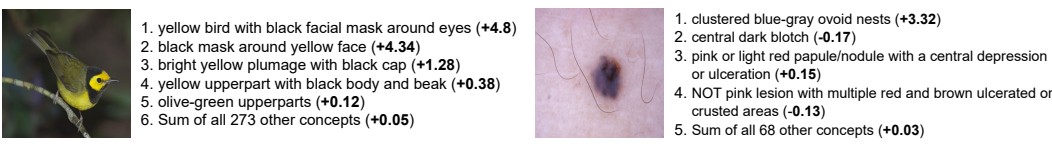

(a) Correctly predicted *Hooded Warbler*      (b) *Melanocytic Nevus* wrongly predicted as *BCC*

Figure 4: Per-image explanations of M-CBM at NCC=5 for a correct prediction in CUB (a) and a misclassification in ISIC 2018 (b). Concepts with negative logit are labeled as "NOT concept".

## 7   CONCLUSION AND LIMITATIONS

We presented Mechanistic Concept Bottleneck Models (M-CBMs), a novel paradigm for training CBMs using concepts learned directly from a black-box backbone and automatically annotated by an MLLM. With this approach, we substantially improve over the state-of-the-art, both in terms of

task accuracy and concept predictions. We are also able to keep explanations concise by controlling final layer sparsity to achieve a target Number of Contributing Concepts (NCC). One limitation general to all CBMs is that we still lack a systematic way to assess whether concepts are learned as intended and not via spurious correlations. This is because the final layer is interpretable, but the concept prediction remains a black-box. Another limitation is that, while NCC allows us to control the accuracy–leakage trade-off, it is still not enough to eliminate leakage, as CBMs trained on random words still achieve much higher accuracy than we would expect from random chance. In future work, it may be interesting to investigate whether adding more bottleneck layers can mitigate this by making it harder for information to leak through. Furthermore, compared to other baselines, M-CBM is less plug-and-play, requiring some supervision to ensure that concepts extracted via SAE are interpretable (see Appendix B), and that the MLLM is providing high-quality annotations. Finally, the high computational cost of using MLLMs for annotations can also be considered a limitation, especially for large datasets. Despite these limitations, given that, due to computational constraints, we annotate only a small subset of images, there might be great potential for improvement with the advancements of MLLMs in both performance and efficiency.

## ACKNOWLEDGMENTS

Antonio De Santis was supported by the Progetto Rocca Doctoral Fellowship for his visit to MIT CSAIL. His doctoral scholarship is funded by the Italian Ministry of University and Research (MUR) under the National Recovery and Resilience Plan (NRRP), by Thales Alenia Space, and by the European Union (EU) under the NextGenerationEU project.

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

## A  APPENDIX OVERVIEW

In the appendix, we provide:

- B.  Details on training SAEs
- C.  Details on random words CBMs
- D.  Summary of CBMs parameters and backbones
- E.  Visualizations of CBL neurons
- F.  More examples of explanations
- G.  Additional experiments on dataset annotation
- H.  Accuracies under NEC
- I.  Visualizing SAE features
- J.  Explanations compared with baselines
- K.  Performance with an open-source MLLM

## B  DETAILS ON TRAINING SAES

In this section, we detail how we trained and evaluated SAEs for concept extraction. We did not find a single hyperparameter configuration that worked uniformly across datasets or backbones. Some dataset-specific adjustments were typically required. Following Bricken et al. (2023), we relied on a mix of quantitative and qualitative proxies to judge whether an SAE was "good enough" for downstream concept use. Specifically, we tracked the following metrics:

1. **L2 reconstruction loss**. We want the reconstruction loss to be low to ensure we extract a comprehensive set of concepts.

2. **Average $\ell_0$**. We aim for a number significantly lower than the backbone dimensionality to ensure concepts are disentangled.

3. **Feature density histogram**. It shows how many hidden neurons fire at different activation frequencies across the training set. In the ideal scenario, neurons are either dead or represent interpretable concepts, so we look for a histogram with two clusters, one with very low density representing dead or noisy features and one with higher density, which should represent the actual concepts.

4. **Recovered cross-entropy loss**. We ideally want the extracted concepts to recover model performance, so we know they have predictive power.

5. **Recovered accuracy**. Same as 4. For ISIC2018, we also consider balanced accuracy.

6. **Manual inspection**. Inspecting top-activating images for random neurons in the high-density cluster to assess whether the learned concepts seem interpretable. Empirically, we found that when all other metrics are healthy, most concepts are interpretable, although this cannot be guaranteed.

In Table 3, we provide the training hyperparameters for the SAEs used in the paper, while in Table 4, we show the results in terms of the evaluation metrics we monitored. In Figure 5, we provide the Feature Density Histograms for each SAE. We can see that low-density neurons are generally well separated from high-density neurons. Furthermore, most of the low-density neurons are dead, i.e., never activating. Some neurons are neither dead nor high-density, and these are typically noisy and not very important for the task. As explained in Section 3, we perform a filtering step to remove these neurons before naming and annotation. In Figure 6, we show how choosing a different feature density cut-off impacts recovered loss, accuracy, and the number of neurons kept. We highlight the cut-off we used with a red star symbol. As we see, removing neurons with very low density has little impact on cross-entropy loss and accuracy. After pruning, recovered loss and accuracy for CUB become 89.40% and 98.41%. For ISIC2018, recovered loss and balanced accuracy become 99.41% and 96.84%. For ImageNet, recovered loss and accuracy become 97.63% and 96.60%.

Table 3: Training hyperparameters for the SAEs used in the paper.

| Hyperparameter | CUB | ISIC2018 | ImageNet |
|---|---|---|---|
| Backbone layer dimension | 512 | 2048 | 2048 |
| Expansion factor | $1\times$ | $0.25\times$ | $4\times$ |
| Optimizer | Adam | Adam | Adam |
| Learning rate | $1 \times 10^{-4}$ | $1 \times 10^{-4}$ | $1 \times 10^{-3}$ |
| L1 coefficient ($\lambda_{\text{SAE}}$) | $2 \times 10^{-3}$ | $5 \times 10^{-4}$ | $1 \times 10^{-3}$ |
| Epochs | 1000 | 1000 | 1000 |
| Patience for early stopping | 50 | 50 | 50 |

Table 4: Evaluation metrics for the SAEs used in the concept extraction phase (pre-pruning). These are computed on the validation set, except for $\ell_0$, which is computed on the training set.

| Metric | CUB | ISIC2018 | ImageNet |
|---|---|---|---|
| L2 reconstruction loss | 0.0231 | 0.0066 | 0.0462 |
| Average $\ell_0$ | 7.66 | 17.14 | 39.23 |
| Recovered loss (CE) | 89.49% | 99.58% | 97.74% |
| Recovered accuracy | 98.39% | acc: 96.08%, bal. acc: 96.84% | 96.68% |

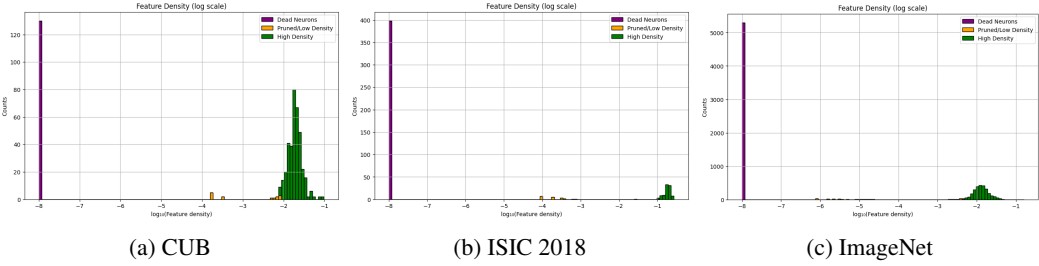

Figure 5: Feature density histogram for CUB, ISIC2018 and ImageNet. Purple indicates dead neurons (never active in the training set). Yellow indicates neurons that were pruned due to low density and little to no impact on recovered loss. Green indicates neurons that are kept as concepts for the subsequent steps.

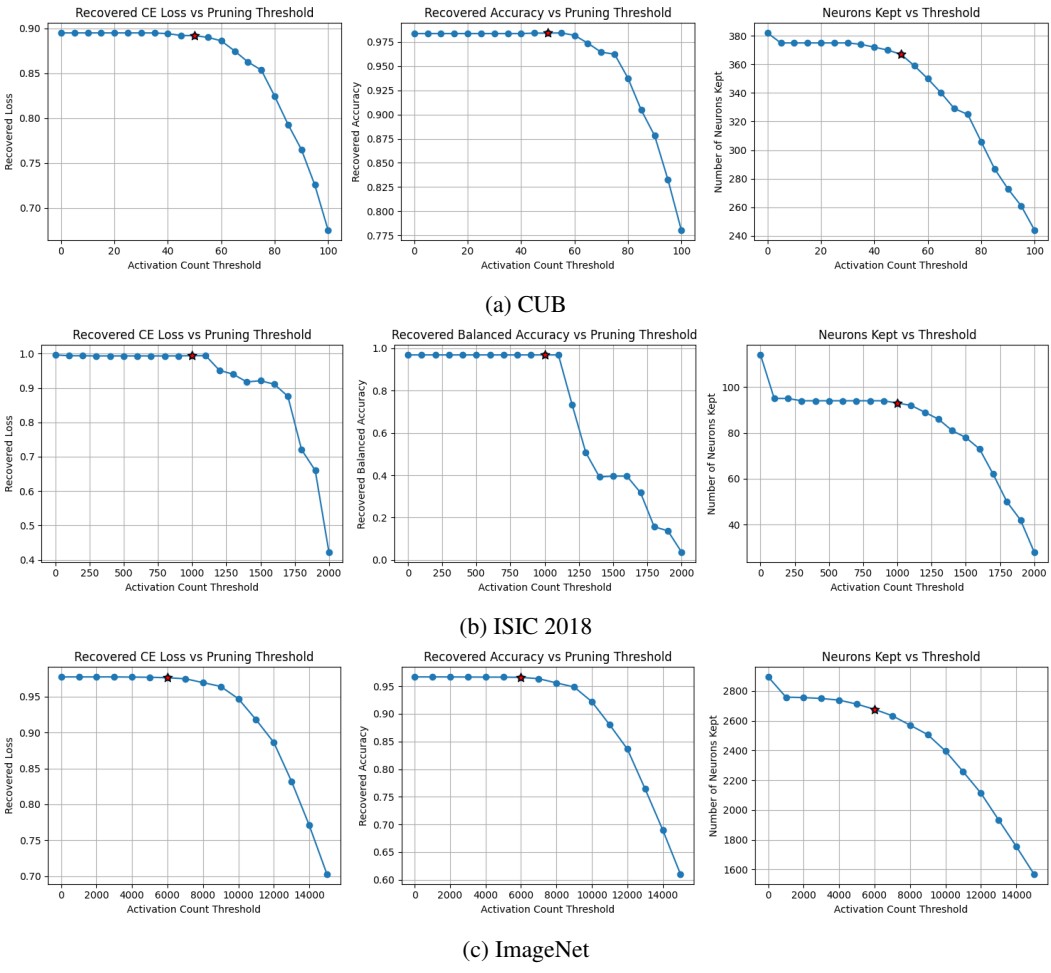

Figure 6: Effect of pruning by activation-count threshold for CUB, ISIC2018, and ImageNet. Higher thresholds typically reduce recovered performance, but when discarding low-density neurons, the reduction tends to be negligible. The point highlighted by a red star indicates the cutoff used in our experiments.

## C DETAILS ON RANDOM WORDS CBMS

In this section, we provide additional details on how we implemented the experiments with random words for VLG-CBM, VLG-CBM$_{CA}$, and M-CBM. For each method, we replace every concept

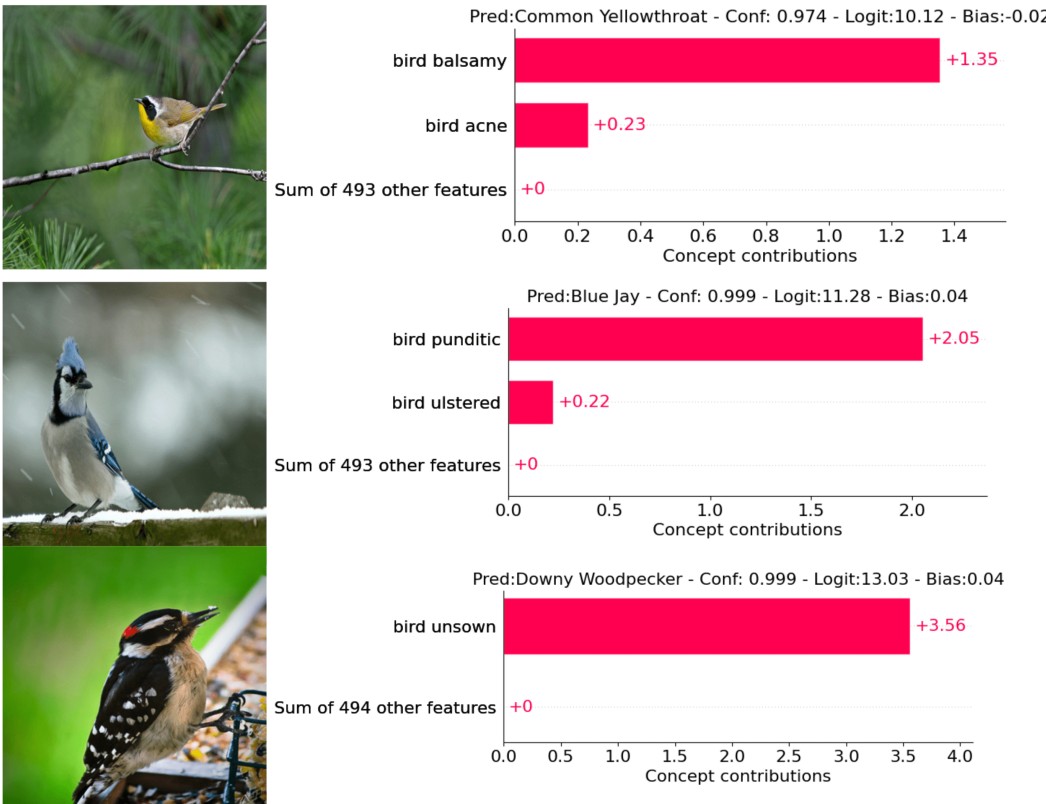

Figure 7: Examples of VLG-CBM explanations where replacing all concept names with random ones still yields correct predictions using just 1–2 concepts, illustrating how class-conditioned annotation can leak class-specific information unrelated to concept semantics.

name with a random, semantically meaningless text while preserving the cardinality of the original concept sets and their class-conditioned assignment for vanilla VLG-CBM. We draw words from the NLTK's words corpus, filtered to lowercase alphabetic strings of length 3–8, and added the prefix "`bird `" so that the result is a short phrase like "`bird pizza`". The prefix helps maintain minimum image relevance so that models like GroundingDINO or GPT4.1 are more likely to annotate the random concepts as positive in some of the images. For VLG-CBM, the annotation is done using their official codebase without modifications, while for VLG-CBM$_{CA}$, we remove class conditioning and annotate each (random) concept across all images. Because GroundingDINO accepts at most 256 input tokens, we batch concept lists and run multiple passes until all concepts are processed. For M-CBM, we follow our standard pipeline but substitute random names before annotation. Furthermore, when annotating random concepts, we omit reference grids of top-activating images to avoid leaking information about the original concepts. Figure 7 illustrates how, under class-conditioned annotation, VLG-CBM can effortlessly predict correctly using only 1–2 random concepts.

## D SUMMARY OF CBMS PARAMETERS AND BACKBONES

In this section, we report the number of parameters used at inference time for all methods and datasets. For each configuration, we decompose the total parameter count into (i) the pre-trained backbone, and (ii) the CBL plus the final classifier, which we denote as CBM. The CBL size $K$ corresponds to the dimensionality of the concept bottleneck (i.e., the total number of concepts). It differs across methods, because each CBM constructs and filters its concept set using its own procedure. Table 5 summarizes the parameter counts (in millions) for all methods considered in our main experiments. Differences are largely dominated by the choice of backbone (e.g., CLIP ViT-B/16 vs ResNet-18/50), while the additional parameters introduced by the CBM head are relatively small in all cases.

Table 5: Inference-time parameter counts (in millions). Backbone counts include only the pretrained feature extractor. CBM counts include the CBL and the final classifier.

| Method | Dataset | Backbone | Backbone (M) | CBL $K$ | CBM (M) | Total (M) |
|---|---|---|---|---|---|---|
| LF-CBM | CUB | ResNet-18 | 11.69 | 208 | 0.15 | 11.84 |
| LF-CBM | ISIC2018 | ResNet-50 | 25.56 | 35 | 0.07 | 25.63 |
| LF-CBM | ImageNet | ResNet-50 | 25.56 | 4523 | 13.79 | 39.35 |
| DN-CBM$_{RN}$ | CUB | CLIP RN50 | 38.30 | 8192 | 10.04 | 48.34 |
| DN-CBM$_{RN}$ | ISIC2018 | CLIP RN50 | 38.30 | 8192 | 8.45 | 46.75 |
| DN-CBM$_{RN}$ | ImageNet | CLIP RN50 | 38.30 | 8192 | 16.59 | 54.89 |
| DN-CBM$_{ViT}$ | CUB | CLIP ViT-B/16 | 86.20 | 4096 | 2.92 | 89.12 |
| DN-CBM$_{ViT}$ | ISIC2018 | CLIP ViT-B/16 | 86.20 | 4096 | 2.13 | 88.33 |
| DN-CBM$_{ViT}$ | ImageNet | CLIP ViT-B/16 | 86.20 | 4096 | 6.20 | 92.40 |
| VLG-CBM$_{CA}$ | CUB | ResNet-18 | 11.69 | 535 | 0.38 | 12.07 |
| VLG-CBM$_{CA}$ | ISIC2018 | ResNet-50 | 25.56 | 80 | 0.16 | 25.72 |
| **M-CBM (Ours)** | CUB | ResNet-18 | **11.69** | **278** | **0.20** | **11.89** |
| **M-CBM (Ours)** | ISIC2018 | ResNet-50 | **25.56** | **73** | **0.15** | **25.71** |
| **M-CBM (Ours)** | ImageNet | ResNet-50 | **25.56** | **2648** | **8.07** | **33.63** |

# E    VISUALIZATIONS OF CBL NEURONS

In Figures 8, 9, and 10 we show the top–5 activating test images for representative concepts on CUB, ISIC2018, and ImageNet, respectively. These visualizations qualitatively assess whether CBL concepts align with their intended semantics and, when paired with model explanations, help convey *what* the model is actually seeing in the image that influences a prediction.

# F    MORE EXAMPLES OF EXPLANATIONS

In this section, we provide additional examples of local explanations of our M-CBMs at NCC=5. The explanations are shown in Figures 11, 12, and 13 respectively from the CUB, ISIC2018, and ImageNet test sets.

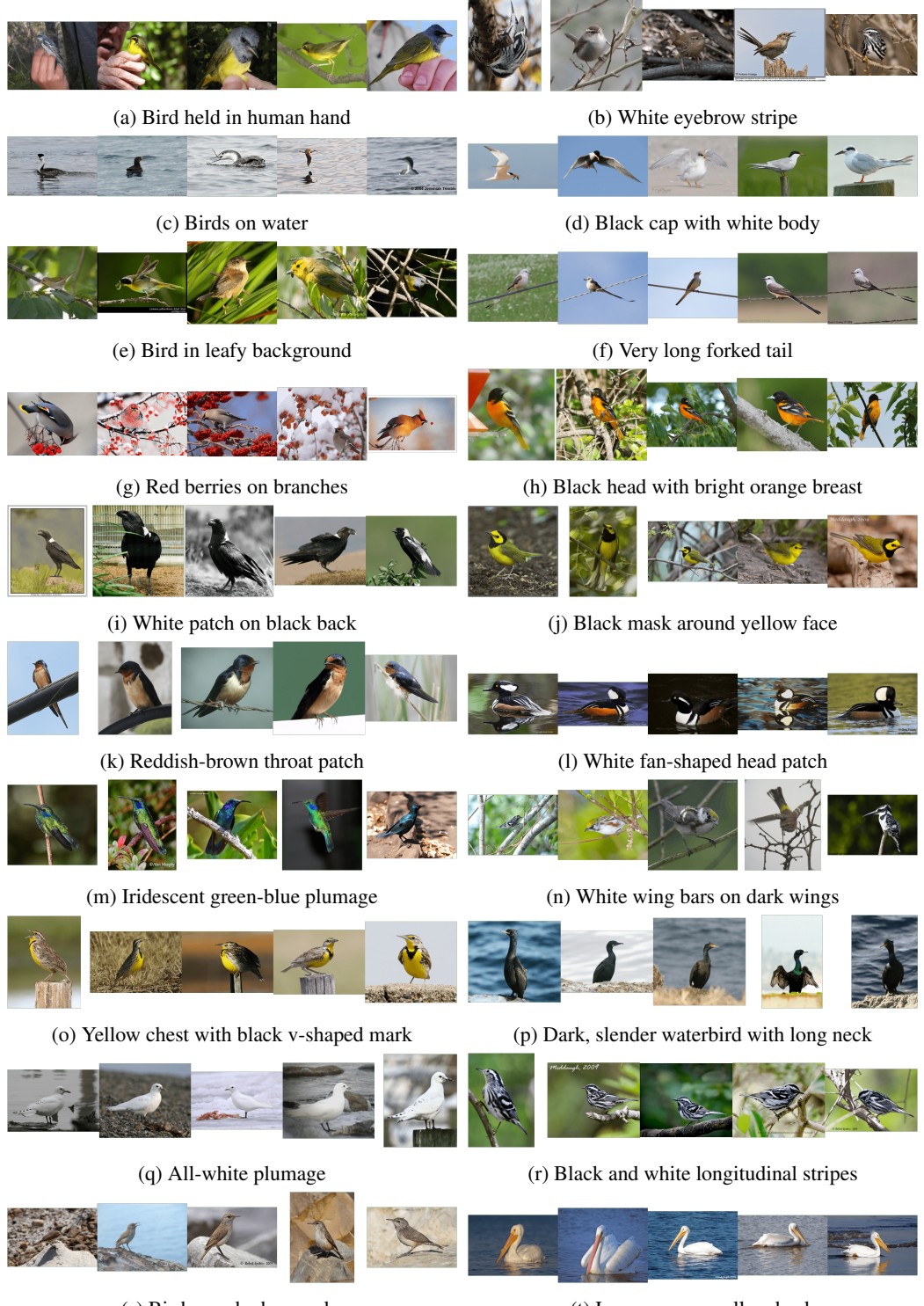

Figure 8: Top-5 activating images for CUB concepts

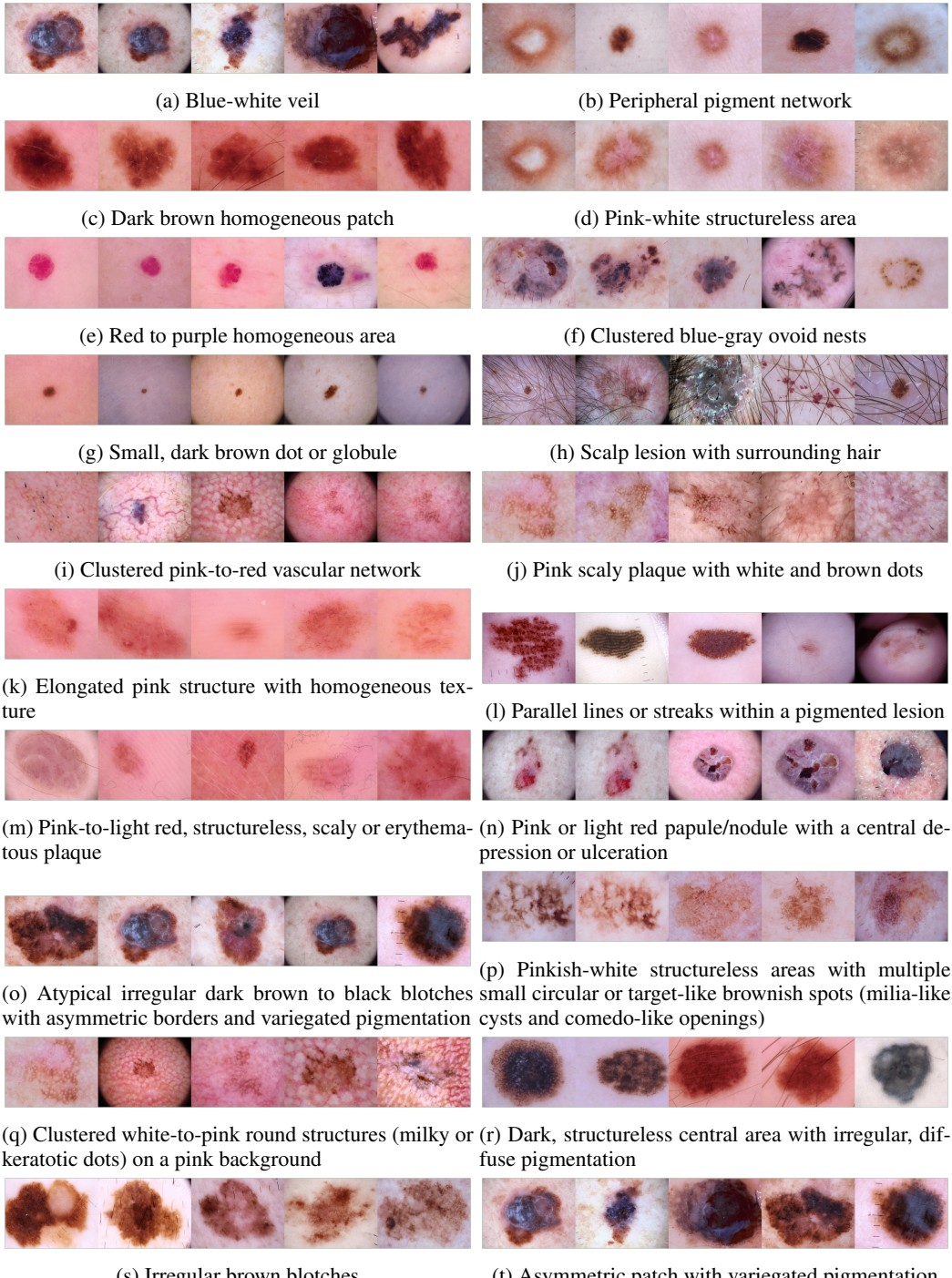

(a) Blue-white veil

(b) Peripheral pigment network

(c) Dark brown homogeneous patch

(d) Pink-white structureless area

(e) Red to purple homogeneous area

(f) Clustered blue-gray ovoid nests

(g) Small, dark brown dot or globule

(h) Scalp lesion with surrounding hair

(i) Clustered pink-to-red vascular network

(j) Pink scaly plaque with white and brown dots

(k) Elongated pink structure with homogeneous texture

(l) Parallel lines or streaks within a pigmented lesion

(m) Pink-to-light red, structureless, scaly or erythematous plaque

(n) Pink or light red papule/nodule with a central depression or ulceration

(o) Atypical irregular dark brown to black blotches with asymmetric borders and variegated pigmentation

(p) Pinkish-white structureless areas with multiple small circular or target-like brownish spots (milia-like cysts and comedo-like openings)

(q) Clustered white-to-pink round structures (milky or keratotic dots) on a pink background

(r) Dark, structureless central area with irregular, diffuse pigmentation

(s) Irregular brown blotches

(t) Asymmetric patch with variegated pigmentation

Figure 9: Top-5 activating images for ISIC2018 concepts

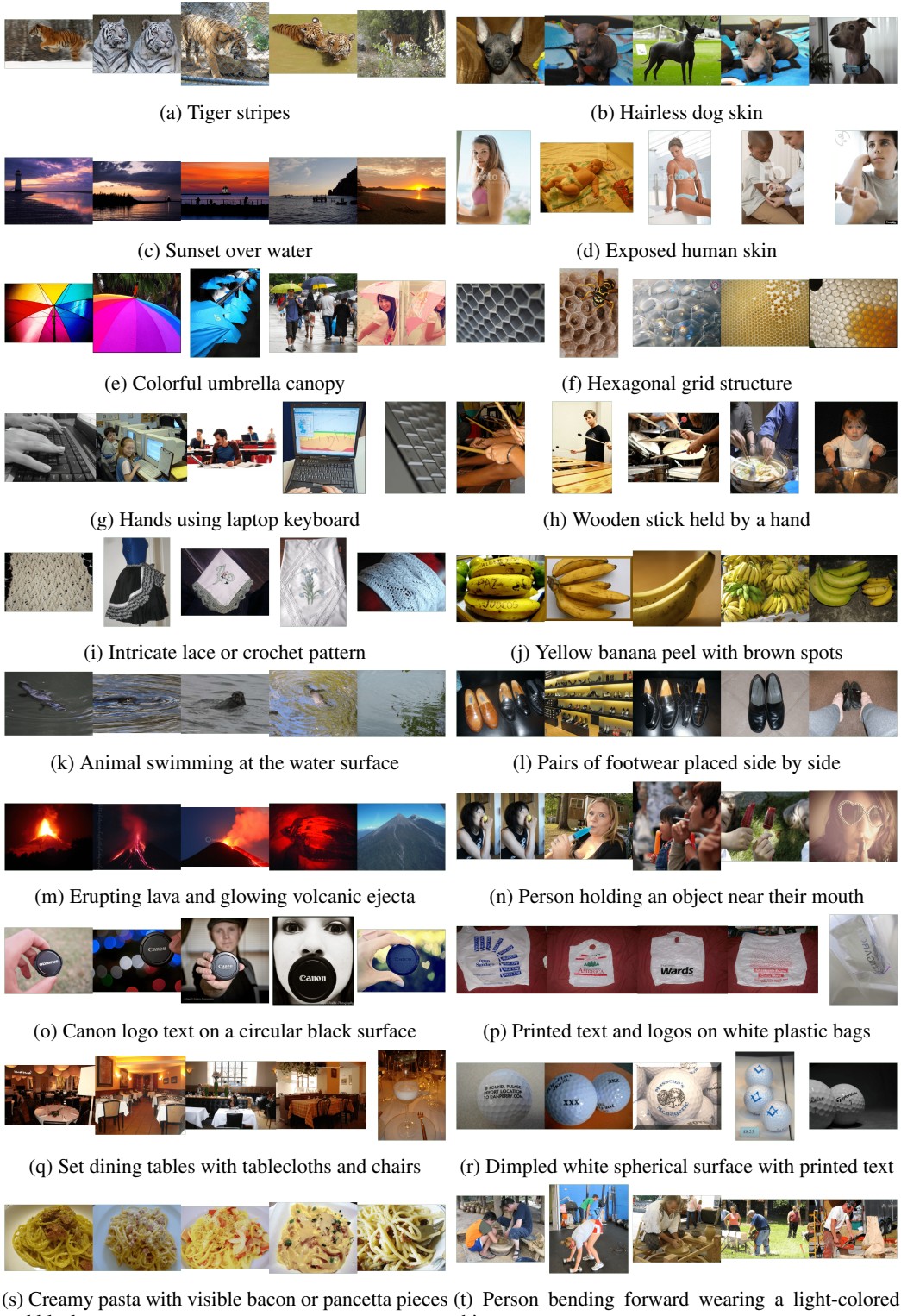

Figure 10: Top-5 activating images for ImageNet concepts

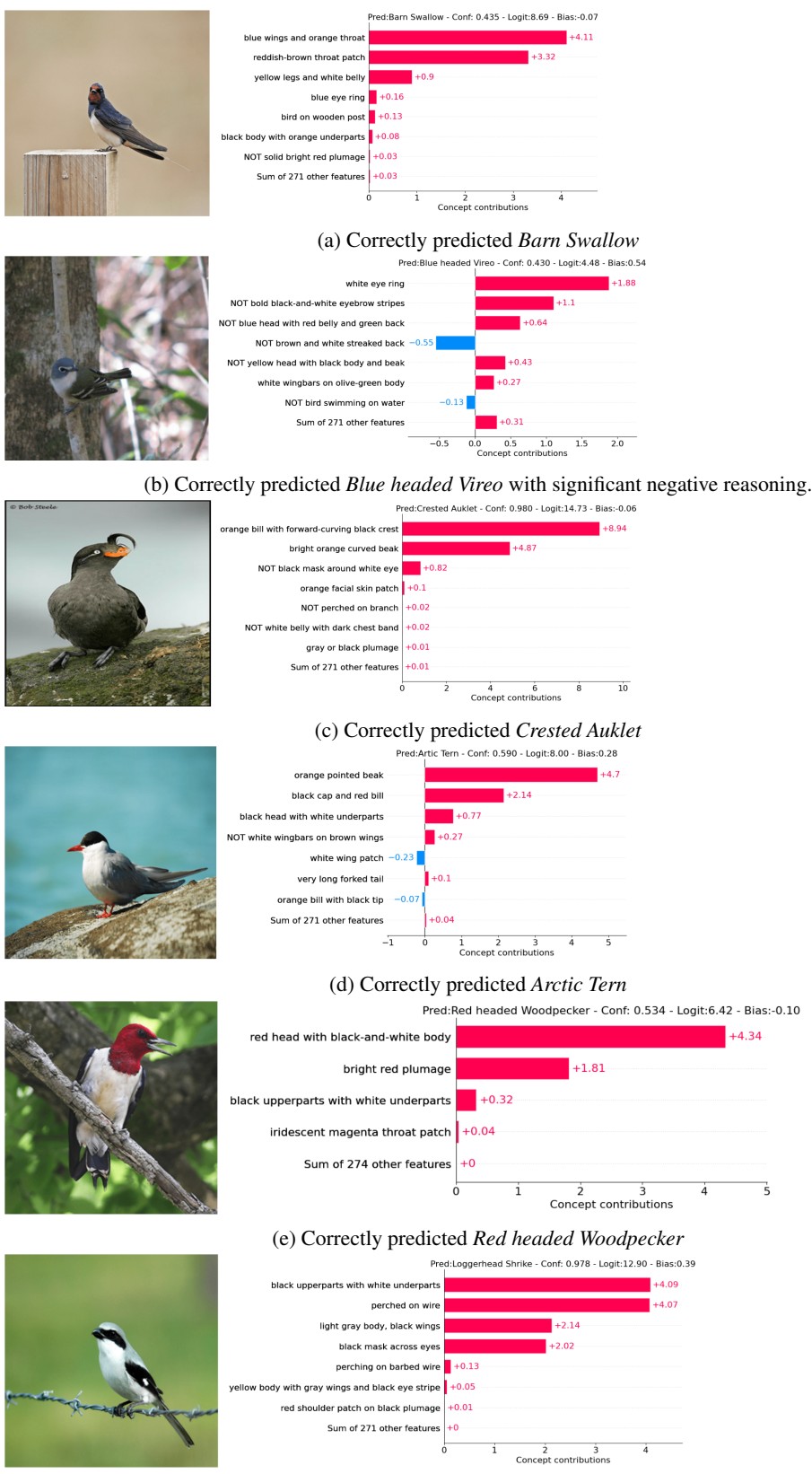

(a) Correctly predicted *Barn Swallow*

(b) Correctly predicted *Blue headed Vireo* with significant negative reasoning.

(c) Correctly predicted *Crested Auklet*

(d) Correctly predicted *Arctic Tern*

(e) Correctly predicted *Red headed Woodpecker*

(f) Correctly predicted *Loggerhead Shrike*

Figure 11: Examples of local explanations from CUB test set.

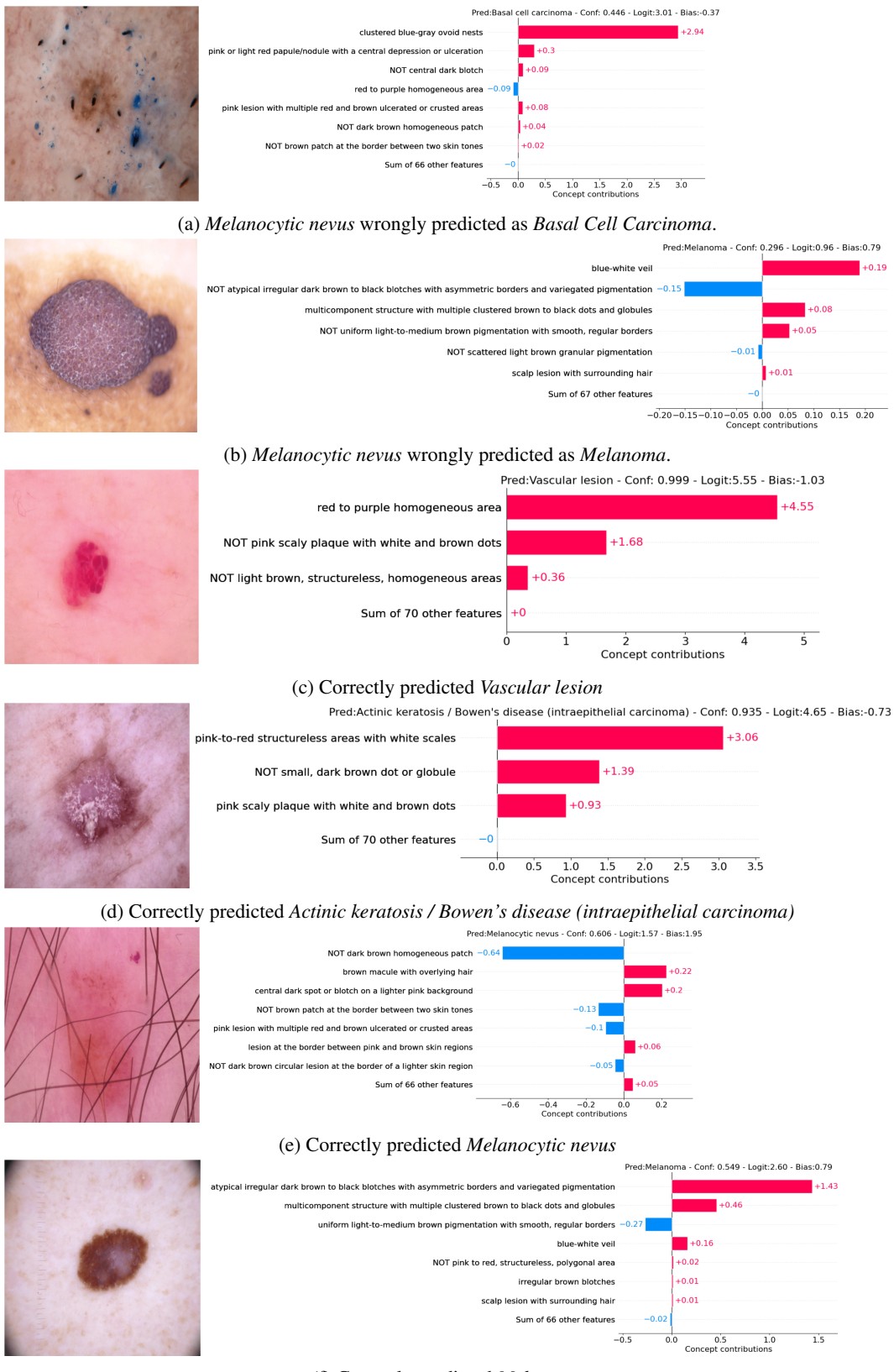

(a) *Melanocytic nevus* wrongly predicted as *Basal Cell Carcinoma*.

(b) *Melanocytic nevus* wrongly predicted as *Melanoma*.

(c) Correctly predicted *Vascular lesion*

(d) Correctly predicted *Actinic keratosis / Bowen's disease (intraepithelial carcinoma)*

(e) Correctly predicted *Melanocytic nevus*

(f) Correctly predicted *Melanoma*

Figure 12: Examples of local explanations from ISIC2018 test set.

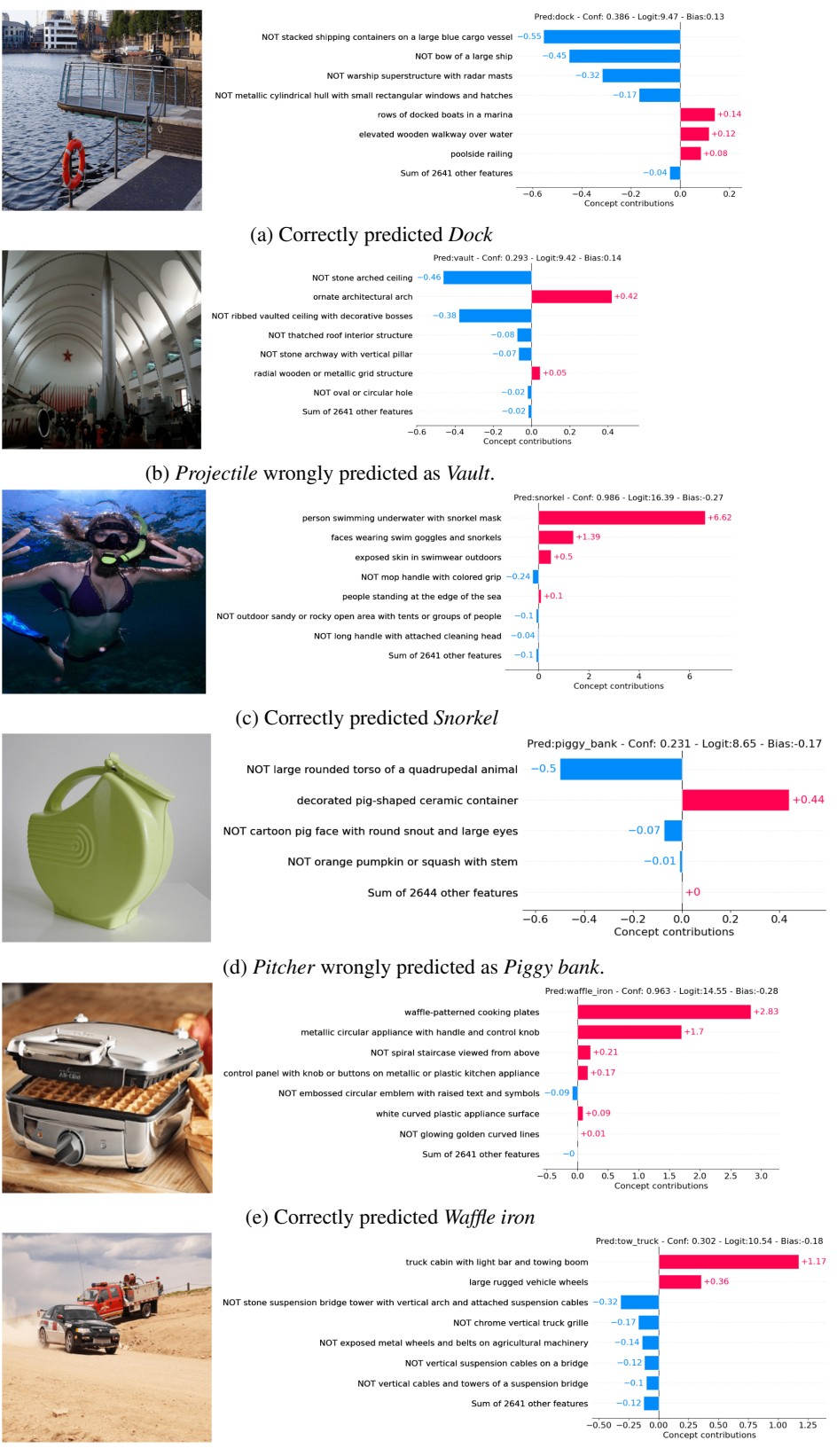

Figure 13: Examples of local explanations from ImageNet test set.

## G    ADDITIONAL EXPERIMENTS ON DATASET ANNOTATION

**On the effect of the reference set for annotation quality.**    Our annotation pipeline provides the MLLM with a reference set of active examples for the target concept before asking it to annotate new samples. The goal is to visually ground the concept, which can be especially important when the textual label alone may not be precise enough (e.g., long tail, rufous crown, or small nevus). To quantitatively assess how much this reference set contributes to annotation quality, we use the human-annotated concept labels available for the CUB dataset and compare the MLLM's performance with and without providing the reference images. For each concept, we run two parallel annotations over the same set of randomly selected 25 images: (i) one where the model is first shown 25 positive examples of the concept, and (ii) one where the model sees only the concept name and the images to annotate. Ground-truth concept labels from CUB allow us to compute accuracy in each condition. Of the 312 CUB attributes, we exclude 19 concepts that contain fewer than 50 positive examples, leaving 293 concepts for evaluation for a total of 7325 images to be annotated. Across these concepts, providing the reference grid achieves a measurable improvement as accuracy increases from 65.91% (without reference) to 69.46% (with reference images). This shows that the MLLM effectively leverages the visual examples for better annotation. However, it is important to note that these numbers may not be representative of the absolute annotation quality, given that CUB concept annotations are known to be very noisy. As noted by Koh et al. (2020), each attribute was annotated by a single non-expert crowdworker, and many concepts are fine-grained or subjective (e.g., red belly vs rufous belly). While such inconsistencies limit the maximum attainable accuracy, the relative improvement provides evidence in favor of the reference set being helpful in improving the quality of the annotations.

**Grid vs single-image for dataset annotation.**    In the main annotation pipeline, we present the MLLM with a $5 \times 5$ grid of 25 images at once, asking it to output a list of binary predictions corresponding to each image. This batch style strategy is critical for keeping annotation costs manageable, especially for large datasets such as ImageNet. However, it is natural to ask whether the MLLM would produce higher-quality annotations if instead queried with one image at a time. To investigate this, we repeat the experiment described in the previous section, but modify the prompting so that each image is shown individually. For every concept, the model receives the same visual reference examples, but is presented with each of the 25 test images in separate calls. We compare the resulting accuracy with the grid-based setup.

The single image approach has an accuracy of 74.00%, compared to 69.46% for the grid. This indicates that the MLLM indeed benefits from focusing on one image at a time, likely because it reduces cross-image interference and allows the model to allocate all attention to the visual details of a single example. In principle, this also suggests that prompting granularity is an important design dimension in MLLM-based annotation systems. However, the practical implications of this improvement are highly limited. The single-image method increases the number of model calls by a factor of 25, as each of the 25 images in the original batch now requires its own MLLM request. As an example, for ImageNet this would increase annotation time from days to months and the API cost from USD 370 to USD 9250. As a result, while the single-image approach achieves better annotations, it is currently impractical for most applications. Nevertheless, these findings are encouraging, as they indicate that as MLLMs become more cost-efficient, single-image prompting would be a straightforward way to improve results further.

**Robustness of MLLM annotation under reference set poisoning.**    Providing a set of reference images for a target concept generally improves annotation quality, but a potential risk is that if this reference set is noisy (i.e., containing some images that represent different features than the concept), the MLLM might inadvertently internalize these spurious visual features and associate them with the concept. To test whether our annotation pipeline is susceptible to this effect, we design a controlled poisoning experiment in which the reference images for a concept are deliberately contaminated with images from a different concept. We then also contaminate the batch of images to be annotated with images from this secondary concept and measure whether these poisoned images are predicted as positive more often than typical negative examples. This setup provides a direct quantitative assessment of whether the MLLM forms spurious visual associations from the reference examples or remains faithful to the intended concept semantics. The experiment is carried out on the CUB dataset, where ground-truth concept annotations allow measurement of false positive behavior.

For each target concept $A$, we construct two types of negative examples: (i) *standard negatives* (images that do not contain $A$) and (ii) *poisoned negatives* (images that do not contain $A$ but do contain a different concept $B$). For every concept $A$, we randomly select a concept $B$ and contaminate its reference set by replacing 5 out of the 25 reference images with poisoned negatives. Then, during annotation, we also replace 1 standard negative image per annotation batch with a poisoned negative. The remaining pipeline is kept unchanged. We repeat this procedure for all 293 concepts with sufficiently many positive samples and annotate 25 images per concept, resulting in 293 MLLM annotation calls and a total of 7325 images.

After annotation, we compute the False Positive Rate (FPR) separately for poisoned negatives and for standard negatives, quantify the difference using Cohen's $h$ effect size, and test whether the two proportions differ significantly using a two-proportion $z$-test. Results are reported in Table 6. The results reveal only a small difference between the FPR of poisoned and normal negatives: 0.232 vs 0.198. The corresponding effect size, Cohen's $h = 0.084$, is well below the conventional threshold for a "small" effect ($h = 0.2$), indicating that the magnitude of this difference is tiny and negligible in practical terms. Consistently, the two-proportion $z$-test does not find a statistically significant difference between the two FPRs ($p$-value $= 0.157$), meaning that the observed effect is statistically indistinguishable from random variation. Taken together, these findings suggest that the MLLM does not meaningfully internalize the spurious visual features introduced through reference-set poisoning. Even when 20% of the reference examples are deliberately contaminated with a conflicting concept, the model's annotation behavior remains largely stable, indicating that the annotation is primarily guided by the semantic description of the target concept rather than incidental correlations present in the reference examples.

Table 6: Results of MLLM-based concept annotation under poisoning of the reference set on the CUB dataset using human-annotated concepts as ground truth. We report the false positive rates (FPR) for poisoned vs normal negative examples, the two-proportion $z$-test, and the effect size (Cohen's $h$).

| Metric | Value |
|---|---|
| FPR (poisoned) | 0.232 |
| FPR (normal) | 0.198 |
| Two-proportion $z$-test ($p$-value) | 0.157 |
| Cohen's $h$ effect size | 0.084 |

## H  ACCURACIES UNDER NEC

In this section, we report M-CBM accuracies when controlling for NEC instead of NCC. For context, NEC measures the average number of non-zero classifier weights per class (e.g., NEC=5 means that, on average, 5 concepts per class have non-zero weights), while NCC is defined in detail in Section 4 and intuitively measures how many concepts, on average, are needed to explain predictions. NCC can be viewed as a generalization of NEC, as when the coverage level is set to $\tau = 1$, NCC reduces to NEC (see Section H.1 for proof). Because we use a relatively high coverage level $\tau = 0.95$, the resulting accuracies are not dramatically different (see Table 7). Intuitively, NCC differs from NEC in that it measures sparsity at the decision-level using concept contributions (weights multiplied by activations), rather than just weights. This relaxes the hard cap that NEC implicitly imposes on the effective concept vocabulary. Indeed, with $NCC_\tau = 5$, the model may assign non-zero weights to

Table 7: Accuracy results at NEC=5 and NEC=avg.

| Dataset | | CUB | | ISIC2018 | | | | ImageNet | |
|---|---|---|---|---|---|---|---|---|---|---|
| Metrics | | Accuracy | | Accuracy | | Balanced Accuracy | | Accuracy | |
| Sparsity | | NEC=5 | NEC=avg | NEC=5 | NEC=avg | NEC=5 | NEC=avg | NEC=5 | NEC=avg |
| M-CBM | | 72.54% | 73.70% | 69.38% | 73.47% | 68.97% | 70.45% | 71.23% | 73.00% |

many concepts, as long as only about 5 concepts are required, on average, to explain each prediction at coverage level $\tau$. For instance, M-CBM at $\text{NCC}_{0.95} = 5$ has 271 concepts with non-zero weights on CUB, 47 on ISIC2018, and 2572 on ImageNet, while at NEC=5 it has 253 on CUB, 29 on ISIC2018, and 2343 on ImageNet. In Figure 14, we provide some examples of concepts that are used by the NCC=5 model but not by the NEC=5 model.

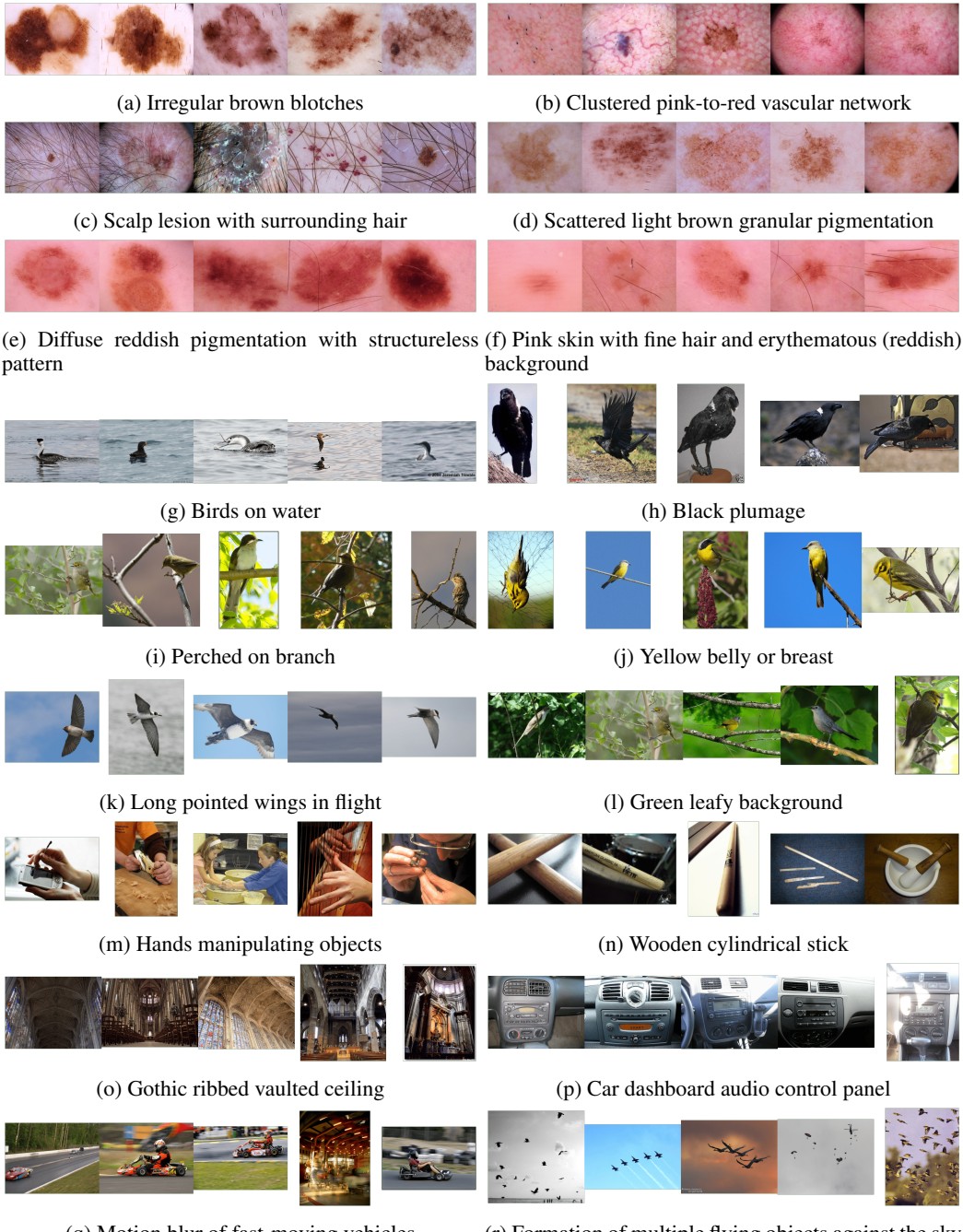

(a) Irregular brown blotches

(b) Clustered pink-to-red vascular network

(c) Scalp lesion with surrounding hair

(d) Scattered light brown granular pigmentation

(e) Diffuse reddish pigmentation with structureless pattern

(f) Pink skin with fine hair and erythematous (reddish) background

(g) Birds on water

(h) Black plumage

(i) Perched on branch

(j) Yellow belly or breast

(k) Long pointed wings in flight

(l) Green leafy background

(m) Hands manipulating objects

(n) Wooden cylindrical stick

(o) Gothic ribbed vaulted ceiling

(p) Car dashboard audio control panel

(q) Motion blur of fast-moving vehicles

(r) Formation of multiple flying objects against the sky

Figure 14: Top-5 activating images for CBL neurons that have all zero weights at NEC=5, but not at NCC=5 for all datasets.

## H.1 RELATIONSHIP BETWEEN NEC AND NCC

We now formalize the connection between NEC and NCC at $\tau = 1$.

**Proposition H.1.** *Let*

$$\text{NEC}(\boldsymbol{W}_F) = \frac{1}{C} \sum_{r=1}^{C} \sum_{k=1}^{K} \mathbf{1}\big\{[\boldsymbol{W}_F]_{k,r} \neq 0\big\},$$

*and let* $\text{NCC}_\tau$ *be defined as in Section 4. Assume that whenever* $[\boldsymbol{W}_F]_{k,r} \neq 0$*, the corresponding concept logit* $[g(\boldsymbol{a}^{(i)})]_k$ *is non-zero for all images* $i$*. This assumption is reasonable, since concept logits are continuous, $z$-normalized values, so the probability of encountering an exact zero is negligible at machine precision. Then*

$$\text{NCC}_1 = \text{NEC}(\boldsymbol{W}_F).$$

*Proof.* By definition in Section 4,

$$u_{k,r}^{(i)} = \big| [g(\boldsymbol{a}^{(i)})]_k \, [\boldsymbol{W}_F]_{k,r} \big| \geq 0.$$

If $[\boldsymbol{W}_F]_{k,r} = 0$, then $u_{k,r}^{(i)} = 0$ for all $i$. Under our assumption, if $[\boldsymbol{W}_F]_{k,r} \neq 0$, then $[g(\boldsymbol{a}^{(i)})]_k \neq 0$ for all $i$, hence $u_{k,r}^{(i)} > 0$ for all $i$. Thus, for fixed $r$,

$$u_{k,r}^{(i)} > 0 \quad \Longleftrightarrow \quad [\boldsymbol{W}_F]_{k,r} \neq 0 \quad \text{for all } i.$$

For each class $r$, define

$$m_r = \sum_{k=1}^{K} \mathbf{1}\big\{[\boldsymbol{W}_F]_{k,r} \neq 0\big\},$$

the number of concepts with non-zero weight for class $r$. For any image $i$, exactly $m_r$ of the $u_{k,r}^{(i)}$ are strictly positive and the remaining $K - m_r$ contributions are zero.

Let $u_{(1),r}^{(i)} \geq \cdots \geq u_{(K),r}^{(i)}$ be the sorted contributions. Sorting moves the $m_r$ positive terms to the front, so

$$\sum_{k=1}^{K} u_{k,r}^{(i)} = \sum_{s=1}^{m_r} u_{(s),r}^{(i)}$$

and $u_{(s),r}^{(i)} = 0$ for all $s > m_r$.

By the definition of $\kappa_r^{(i)}(1)$ in Section 4, it is the smallest number of top-ranked contributions whose sum matches the total contribution $\sum_{k=1}^{K} u_{k,r}^{(i)}$. Since this total is exactly the sum of the $m_r$ strictly positive terms and all remaining terms are zero, it follows that

$$\kappa_r^{(i)}(1) = m_r \quad \text{for all } i, r.$$

Substituting into the definition of $\text{NCC}_1$,

$$\text{NCC}_1 = \frac{1}{|\mathbb{D}| \, C} \sum_{i,r} \kappa_r^{(i)}(1) = \frac{1}{|\mathbb{D}| \, C} \sum_{i,r} m_r = \frac{1}{C} \sum_{r=1}^{C} m_r$$

Using the definition of $m_r$,

$$\text{NCC}_1 = \frac{1}{C} \sum_{r=1}^{C} \sum_{k=1}^{K} \mathbf{1}\big\{[\boldsymbol{W}_F]_{k,r} \neq 0\big\} = \text{NEC}(\boldsymbol{W}_F),$$

which completes the proof. $\square$

# I  Visualizing SAE features

In Figures 15, 17, and 16 we show the top–5 activating images for SAE neurons on CUB, ISIC2018, and ImageNet, respectively. We also provide their respective description given by the MLLM. These visualizations qualitatively assess whether the MLLM descriptions align with the neuron behavior. We also overlay the activating images with their respective saliency map to show where that neuron was looking.

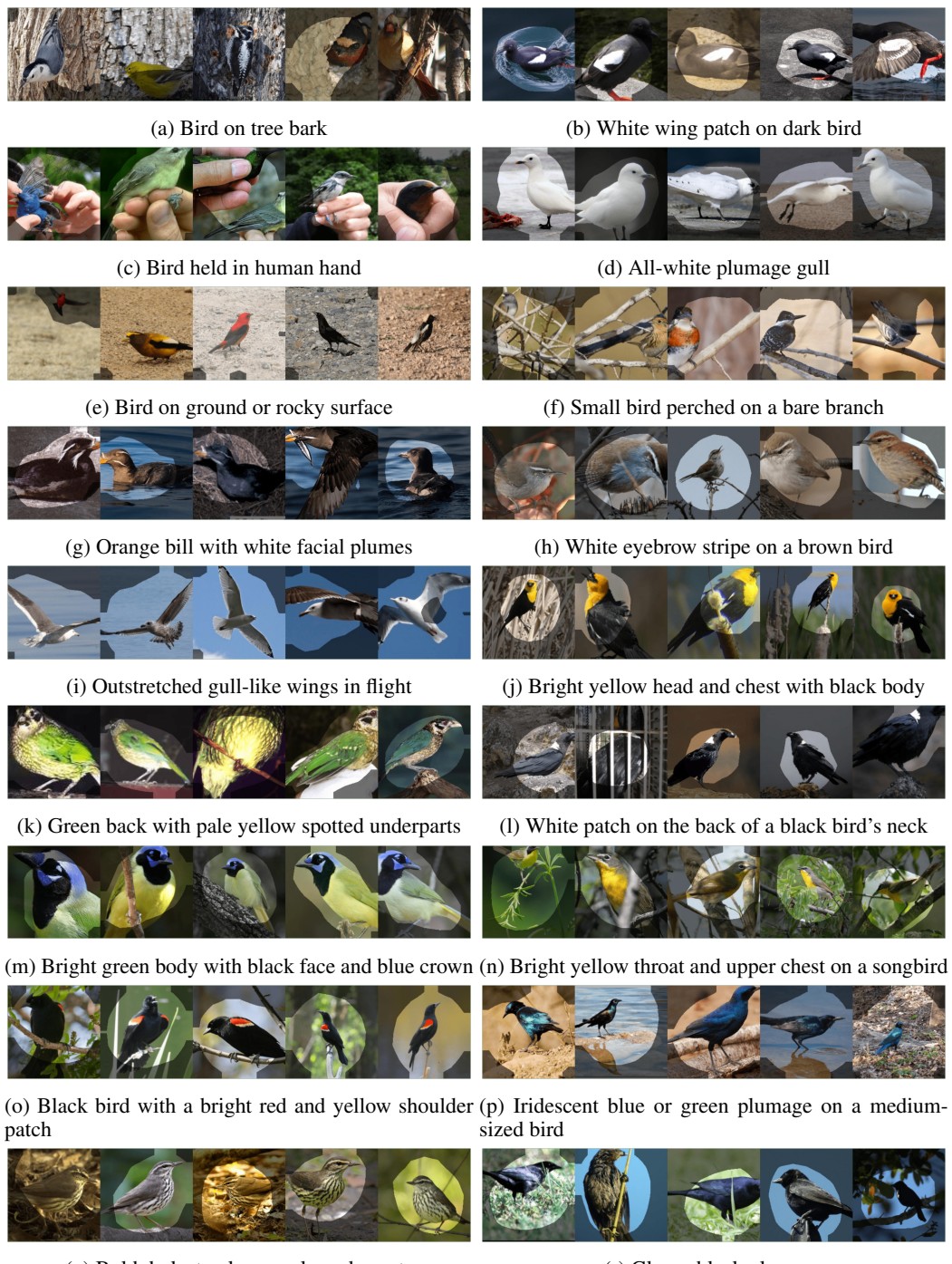

(a) Bird on tree bark

(b) White wing patch on dark bird

(c) Bird held in human hand

(d) All-white plumage gull

(e) Bird on ground or rocky surface

(f) Small bird perched on a bare branch

(g) Orange bill with white facial plumes

(h) White eyebrow stripe on a brown bird

(i) Outstretched gull-like wings in flight

(j) Bright yellow head and chest with black body

(k) Green back with pale yellow spotted underparts

(l) White patch on the back of a black bird's neck

(m) Bright green body with black face and blue crown

(n) Bright yellow throat and upper chest on a songbird

(o) Black bird with a bright red and yellow shoulder patch

(p) Iridescent blue or green plumage on a medium-sized bird

(q) Bold dark streaks on pale underparts

(r) Glossy black plumage

Figure 15: Top-5 activating images for SAE features in CUB

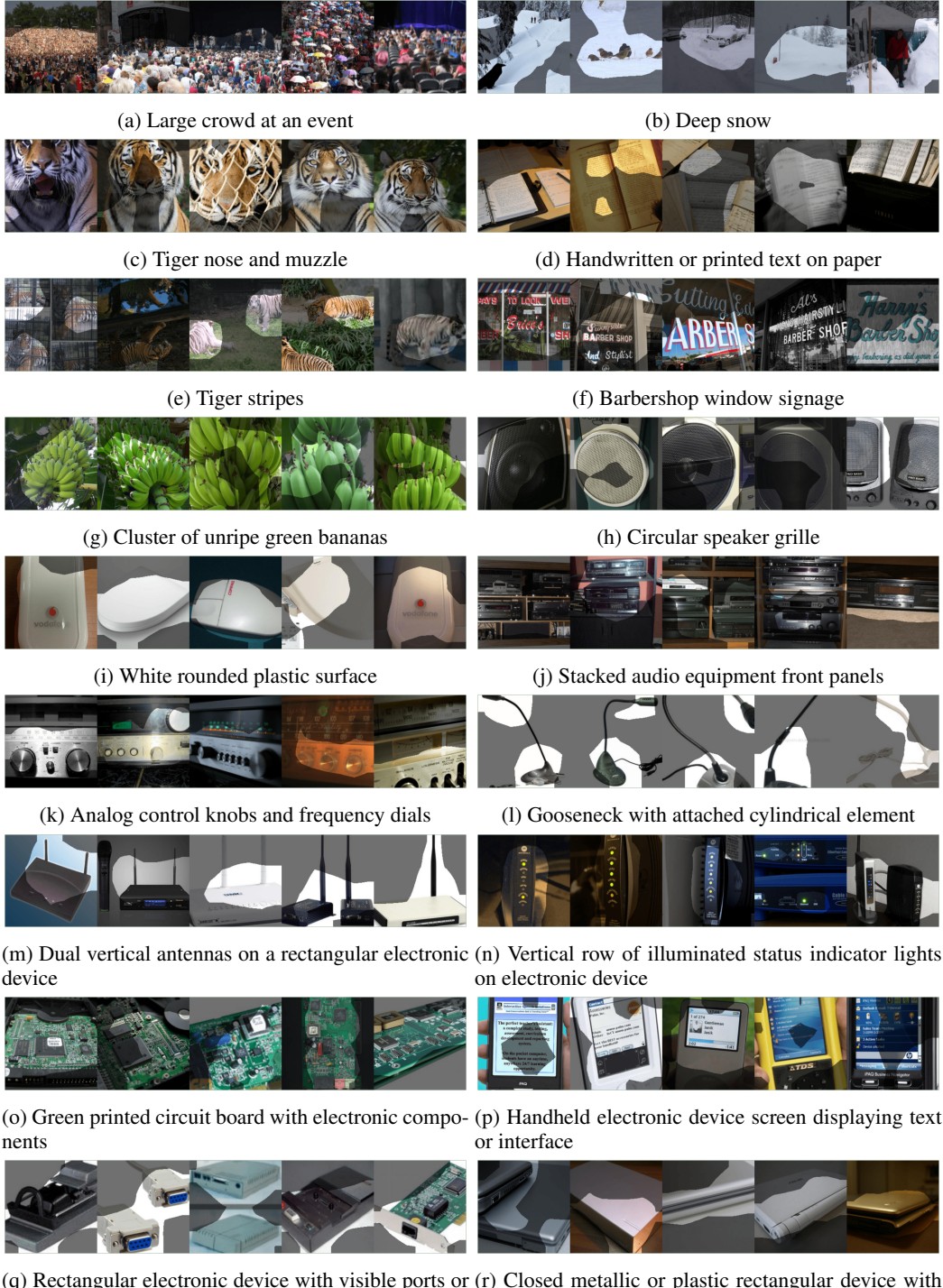

Figure 16: Top-5 activating images for SAE features in ImageNet

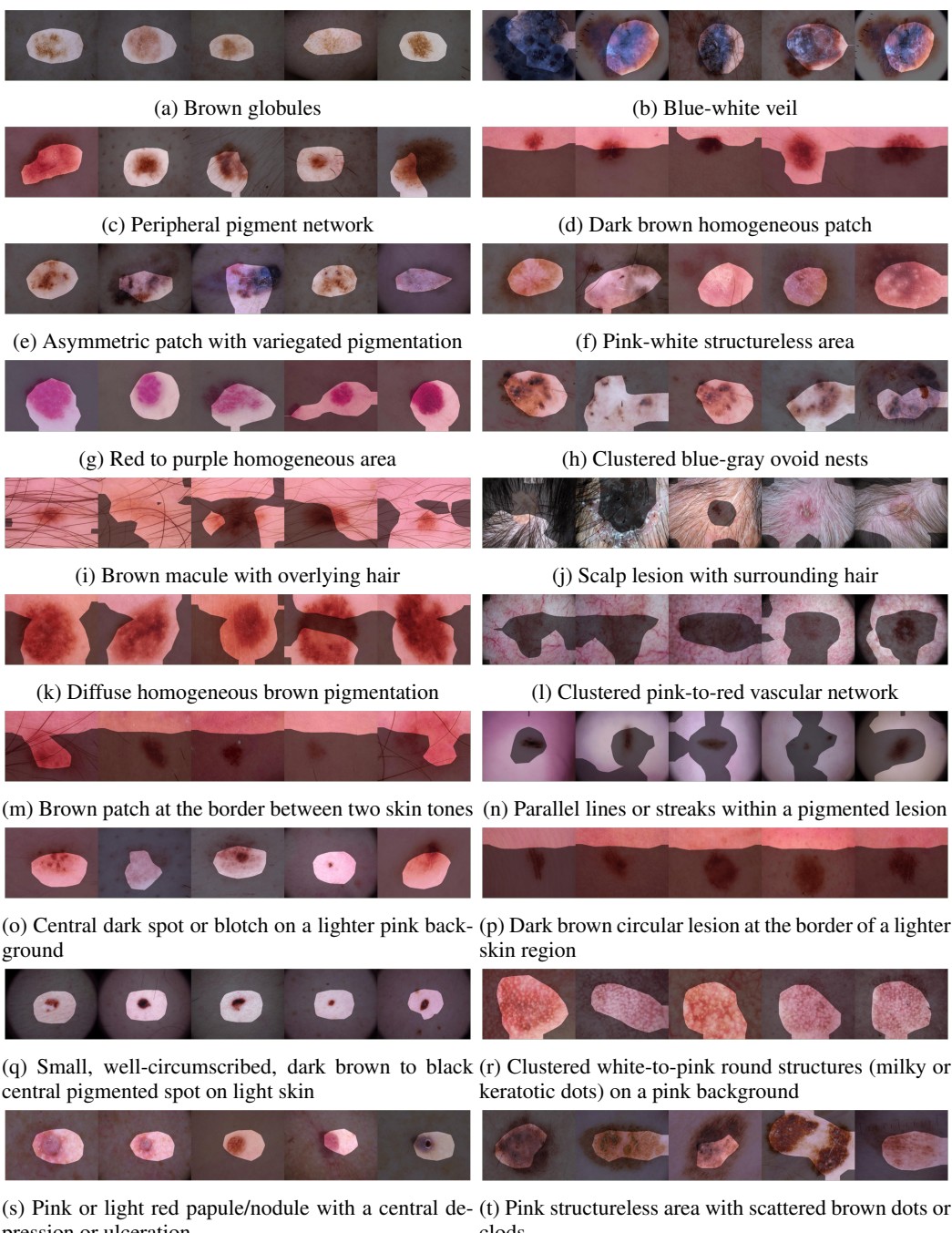

(a) Brown globules

(b) Blue-white veil

(c) Peripheral pigment network

(d) Dark brown homogeneous patch

(e) Asymmetric patch with variegated pigmentation

(f) Pink-white structureless area

(g) Red to purple homogeneous area

(h) Clustered blue-gray ovoid nests

(i) Brown macule with overlying hair

(j) Scalp lesion with surrounding hair

(k) Diffuse homogeneous brown pigmentation

(l) Clustered pink-to-red vascular network

(m) Brown patch at the border between two skin tones

(n) Parallel lines or streaks within a pigmented lesion

(o) Central dark spot or blotch on a lighter pink background

(p) Dark brown circular lesion at the border of a lighter skin region

(q) Small, well-circumscribed, dark brown to black central pigmented spot on light skin

(r) Clustered white-to-pink round structures (milky or keratotic dots) on a pink background

(s) Pink or light red papule/nodule with a central depression or ulceration

(t) Pink structureless area with scattered brown dots or clods

Figure 17: Top-5 activating images for SAE features in ISIC2018

## J    EXPLANATIONS COMPARED WITH BASELINES

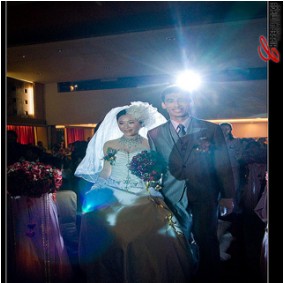

Ground Truth: **Bridegroom**

### M-CBM (Ours)

Predicted: **Bridegroom (Conf. 0.753)**

1. bride in white wedding dress holding bouquet (**+4.07**)
2. group_of_people_in_celebratory_or_formal_attire (**+2.06**)
3. men's suit jacket with dress shirt and necktie (**+1.62**)
4. wedding dress bodice and upper skirt (**+1.27**)
5. standing person in formal or distinctive clothing (**+0.46**)
6. Sum of all 2643 other concepts (+**0.26**)

### DN-CBM$_{ViT}$

Predicted: **Bridegroom (Conf. 0.948)**

1. bridal (**+6.42**)
2. couple (**+2.05**)
3. bejing (**+1.12**)
4. suits (**+0.3**)
5. parties (**+0.17**)
6. Sum of all 4091 other concepts (+**0.1**)

### LF-CBM

Predicted: **Bridegroom (Conf. 0.135)**

1. a wedding (**+4.34**)
2. waiting at the altar (**+0.55**)
3. groom (+**0.35**)
4. Sum of all 4520 other concepts (**0.0**)

Figure 18: ImageNet Dataset

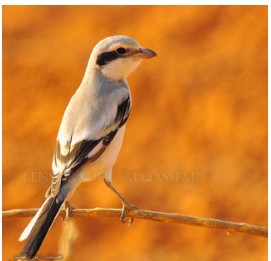

Ground Truth: **Great Grey Shrike**

### M-CBM (Ours)

Predicted: **Great Grey Shrike (Conf. 0.70)**

1. black mask across eyes (**+4.64**)
2. gray body with white underparts (**+1.75**)
3. white wingbars on gray wings (**+1.6**)
4. dark cap with pale cheek (**+0.3**)
5. light gray body, black wings (**+0.23**)
6. Sum of all 273 other concepts (**0.0**)

### VLG-CBM$_{CA}$

Predicted: **Great Grey Shrike (Conf. 0.41)**

1. NOT red shoulders (**+1.8**)
2. NOT stocky body (**+1.35**)
3. NOT large black spider (**+0.88**)
4. NOT small blue green body (**+0.77**)
5. long black and white tail (**+0.6**)
6. Sum of all 530 other concepts (**+0.85**)

### LF-CBM

Predicted: **Great Grey Shrike (Conf. 0.08)**

1. black cap and white "eyeline" (**+1.51**)
2. a white breast (**+1.48**)
3. white and black coloration (**+0.9**)
4. a white face and underparts (**+0.11**)
5. Sum of all 204 other concepts (**0.0**)

### DN-CBM$_{ViT}$

Predicted: **Scissor tailed Flycatcher (Conf. 0.09)**

1. colors (**+0.62**)
2. kramer (**+0.48**)
3. field (**+0.46**)
4. beige (**+0.45**)
5. flea (**+0.42**)
6. Sum of all 4091 other concepts (**+0.05**)

Figure 19: CUB Dataset

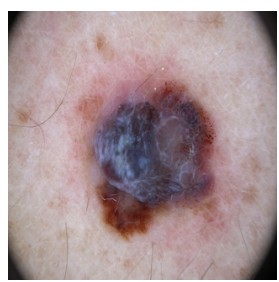

Ground Truth: **Melanoma**

### M-CBM (Ours)

Predicted: **Melanoma (Conf. 0.70)**

1. atypical irregular dark brown to black blotches with asymmetric borders and variegated pigmentation (**+5.0**)
2. NOT uniform light-to-medium brown pigmentation with smooth, regular borders (**+1.03**)
3. blue-white veil (**+0.76**)
4. multicomponent structure with multiple clustered brown to black dots and globules (**+0.75**)
5. NOT scattered light brown pigmentation (-**0.03**)
6. Sum of all 68 other concepts (**0.0**)

### VLG-CBM$_{CA}$

Predicted: **Melanoma (Conf. 0.45)**

1. cancer (**+1.17**)
2. redness erythema (**+0.75**)
3. NOT sun damaged skin (-**0.19**)
4. scaly or rough patches (**+0.14**)
5. NOT scaly or crusted surface (-**0.12**)
6. pathological entity (**+0.08**)
7. ulceration in some cases (**+0.08**)
8. Sum of all 73 other concepts (**+0.05**)

### LF-CBM

Predicted: **Vascular lesion (Conf. 0.54)**

1. NOT dryness or scaling (**+1.16**)
2. NOT hair follicles (**+1.09**)
3. NOT Bleeding (-**0.38**)
4. NOT Scaly or crusted surface (**+0.07**)
5. NOT Epidermis and dermis layers (**+0.01**)
6. Sum of all 30 other concepts (**0.0**)

### DN-CBM$_{ViT}$

Predicted: **Melanoma (Conf. 0.17)**

1. magnesium (**+0.05**)
2. ted (**+0.03**)
3. Sum of all 4094 other concepts (**0.0**)

Figure 20: ISIC Dataset

Table 8: Accuracy results at NCC=5 and NCC=avg of M-CBM using an open-source MLLM (*InternVL3.5-241B-A28B*) for naming and annotation and *e5-large-v2* for concept merging.

| Dataset | CUB | | ISIC2018 | | | | ImageNet | |
|---------|-----|---|----------|---|---|---|----------|---|
| Metrics | Accuracy | | Accuracy | | Balanced Accuracy | | Accuracy | |
| Sparsity | NCC=5 | NCC=avg | NCC=5 | NCC=avg | NCC=5 | NCC=avg | NCC=5 | NCC=avg |
| M-CBM | 71.40% | 74.02% | 67.33% | 74.25% | 69.26% | 72.20% | 56.76% | 68.76% |

Table 9: Evaluation of concept predictions on the test set for M-CBM using an open-source MLLM (*InternVL3.5-241B-A28B*) for naming and annotation and *e5-large-v2* for concept merging.

| Dataset | CUB | | ISIC2018 | | ImageNet | |
|---------|-----|---|----------|---|----------|---|
| Metrics | ROC-AUC | | ROC-AUC | | ROC-AUC | |
| | Macro | Worst-10% | Macro | Worst-10% | Macro | Worst-10% |
| M-CBM | 74.49% | 58.14% | 74.36% | 56.53% | 70.97% | 59.28% |

## K  PERFORMANCE WITH AN OPEN-SOURCE MLLM

In this appendix, we evaluate how M-CBM behaves when replacing the proprietary components used for concept naming, dataset annotation, and concept merging with open-source alternatives while keeping the rest of the pipeline unchanged. For this experiment, we use *InternVL3.5-241B-A28B* as the MLLM for both naming and annotation (it supports multi-image inputs, so we can preserve our prompting protocol), and we use *e5-large-v2* to embed and merge concept names with the same similarity-based merging strategy as in the main paper and the same threshold (i.e., 0.98). Table 8 reports the downstream performance obtained with this fully open-source variant. On CUB, we obtain 71.40% accuracy at NCC=5 and 74.02% at NCC=avg; on ISIC2018, we obtain 67.33% and 74.25% accuracy, with balanced accuracy of 69.26% and 72.20%; and on ImageNet, we obtain 56.76% accuracy at NCC=5 and 68.76% at NCC=avg. Since we do not have ground-truth concept annotations, we follow the same proxy evaluation as in the main paper by annotating the test set with the same pipeline and then measuring ROC-AUC of concept predictions against these labels. Table 9 reports the macro-average ROC-AUC across concepts and the average ROC-AUC over the worst 10% concepts. On CUB, the ROC-AUC is 74.49% (macro) and 58.14% (worst 10%); on ISIC2018, it is 74.36% and 56.53%; and on ImageNet, it is 70.97% and 59.28%. Overall, using the open-source MLLM leads to lower concept-prediction quality and lower downstream performance on all three datasets, but the effect is not uniform across them. On CUB and ISIC2018, the degradation is noticeable but relatively limited, whereas on ImageNet it is substantially larger. Qualitatively, we observe that InternVL3.5 tends to produce more generic and highly similar concept names, which increases the amount of merging and reduces the final concept vocabulary. For example, on CUB we obtain 195 final concepts instead of the 278 concepts produced with GPT-4.1, on ISIC2018 we obtain 38 concepts instead of 73, and on ImageNet we obtain 1768 concepts instead of 2648, again substantially fewer than with GPT-4.1. This effect appears to be particularly problematic on ImageNet, where the much larger and finer-grained set of classes seems to require more concepts and higher-quality annotations. This is also consistent with the lower proxy ROC-AUC values for concept prediction. At the same time, the same qualitative trend of lower annotation quality and fewer final concepts is also present on CUB and ISIC2018, suggesting that concept naming and annotation quality are important across all datasets, even when the downstream impact is smaller. One possible explanation for why the downstream drop is smaller on CUB and ISIC2018 is that these datasets may be more prone to information leakage, since the amount of information needed to separate classes may be substantially smaller than in ImageNet. Overall, these results make even more evident that the quality of concept naming and annotation is critical for downstream classification in M-CBM. On a positive note, since GPT-4.1 annotations are themselves far from perfect and the MLLM component is modular, stronger models may further improve concept quality and, in turn, downstream performance.

