# OpenReview forum: "Learning Concept Bottleneck Models from Mechanistic Explanations"
_ICLR.cc/2026/Conference — ICLR 2026 Poster_

### Official Review · Reviewer_GTFX · 2025-10-21

**Soundness:** 3
**Presentation:** 3
**Contribution:** 2
**Rating:** 4
**Confidence:** 4

**Summary:**

This paper introduces Mechanistic CBM (M-CBM), which introduces advancements in the field of mechanistic interpretability to the concept bottleneck framework. M-CBM takes a pretrained blackbox and trains a sparse autoencoder on its internal activations to extract human-interpretable concepts that are relevant for prediction. After having named and filtered the discovered concepts, a MLLM is used to label a non-random subset of images with these concepts. Subsequently, a standard CBM is trained on the discovered and labelled concepts for human-interpretable predictions. Furthermore, the authors introduce the Number of Contributing Concepts (NCC) metric, which measures the number of concept contributions required to explain $\tau$ of all absolute contributions. Results show improved performance on NCC and concept prediction, as well as qualitative prediction examples.

**Strengths:**

This work proposes to combine the two thriving fields of concept bottleneck models and mechanistic interpretability. As such, it is relevant in the current state of interpretability. It makes use of tested mechanisms from both fields, thereby fostering trust in the proposed method. The method is introduced clearly and the figure aids in understanding. M-CBM also shows NCC improvements compared to relevant baselines. Also, the extensive visualization of discovered concepts and examples in the Appendix is appreciated. Reproducible code is provided too.

**Weaknesses:**

1. While MechInterp is an exciting upcoming field, I am not convinced by the reliability of the concept annotation framework such that there is not a substantial amount of leakage in M-CBM. (i) To name the SAE activations a heatmap is used to localize the concepts. Heatmaps are famously unreliable [1] such that they might not capture what information the blackbox model is using. Then, the highlighted images are passed to the MLLM (Fig. 1), but it is highly unclear to me whether the MLLM truthfully adheres to the highlighted region, and whether the concept interpretation/naming is unique. (ii) It is a good leakage-reducing choice that concepts are subsequently labelled using a MLLM, rather than directly using the SAE activations. Still, there might be a considerable influence from the leaky concept naming critiqued in (i) because "Each call also includes a grid of the top-25 most activating images for the corresponding SAE neuron", which could lead the MLLM to mimic the concept annotation strategy from the leaky SAE, rather than focusing on actual concept presence. Additionally, the MLLM might also use class-leakage where it recognizes an object and determines a concept that relates to the object to be present even if it is not visible in the image.

   To give a concrete example that might ease comprehension, lets say we are interested in the class "Radio". A SAE on a blackbox classifier might find a neuron that always activates when a radio is present. Then a good heatmap highlights most of the radio in each highly activated image. The MLLM that outputs concept names might consequently call the concept "analog control knobs and frequency dials", as it appears in many of the images and is often even highlighted. Note though that the concept name has nothing to do with the actual concept that the model uses, which is "presence of radio". Then, an MLLM is queried to concept annotate a selection of images. Two things that could lead the MLLM to annotate concept presence even if the concept is not there are (a) the 25 highly activated SAE images do not all contain the named concept, thereby the MLLM learns to still assign concept presence for radios even if it is not visible on the image. (b) The MLLM even without context might understand the presence of the object radio and that this object has the aforementioned concept, even if it is not visible, thereby labeling it positive. In the end, the concept labels essentially behave like a class label, but hidden behind an interpretable concept name, which does not accurately capture how it behaves and/or when it activates.

   I am not saying this is necessarily always the case, however, with the current evaluation setup I do not think the amount of this leakage is quantified. Therefore, it is very hard to say whether the extracted concepts (and therefore the method) are meaningful and interpretable.

2. I am not convinced that NEC / NCC are sufficient metrics for capturing information leakage (as stated e.g. in the Abstract). Similar to the previous example, lets say I repurpose a frozen black-box's classifier to be "concept predictor" and rename the classes to some good concept names. Then, initialize the Identity Mapping from my new concept predictor to the classes. As I understand it, this method would have the black-box's accuracy at NCC=1 while being completely uninterpretable and leaking (similar to VLG-CBM). Thus, I do not think the current evaluation setup is able to differentiate meaningful, interpretable posthoc CBMs versus uninterpretable class-leakage.

   To resolve this, off the top of my head, I could think of either performing human user studies or measuring alignment of discovered concepts with known ground truth concepts on a controlled dataset where actual concepts are known.

Other weaknesses:
* In my opinion, the proposed combination of CBM and SAE does not provide a lot of novel ideas but rather combines two existing approaches, which is meaningful, but not original.
* Some concepts in Figure 10, e.g. Tiger stripes or Yellow banana peel with brown spots essentially describe the class, thereby not providing any additional interpretability beyond a blackbox classifier.

[1] Adebayo, Julius, et al. "Sanity checks for saliency maps." Advances in neural information processing systems 31 (2018).

**Questions:**

I invite the authors to address any of my statements from the Weaknesses section, as they are my main source of concern.

For W1:
* Could the authors provide the highly activated images used when querying the MLLM for obtaining the concept names from Figure 3, jointly with the heatmap overlayed, such that it is clear whether concept names relate to the highlighted image parts?
* What is the prompt used for dataset annotation?
* How do the top-25 most activating images affect the MLLM during dataset annotation? Can they be safely removed, or do they affect predictive performance downstream?

Other questions:
* In similar spirit to above, overlaying the heatmaps in Fig. 8-10 would provide additional informativeness.
* Was the consistency of dataset annotation for the 5x5 grid analyzed? I.e. when processing each image separately, would the annotations change? Because as far as I know, MLLMs are not the best at localizing inputs, therefore I am unsure whether a 5x5 grid might pose difficulties.
* Is there a specific reason why elasticnet with $\alpha=0.99$ was chosen instead of just using the simpler LASSO (i.e. $\alpha=1$).

---

> ### Author Response · Authors · 2025-11-21
> **Response to Reviewer GTFX (Part 1)**
>
> Thank you for taking the time to review our paper.
>
> W1. On the reliability of saliency maps, the cited paper studies the (un)reliability of gradient-based saliency maps such as Grad-CAM or Integrated Gradients, which are known to suffer from gradient saturation and baseline selection. Our heatmap is just a weighted sum of activation maps using SAE decoder as weights with no gradients involved. While we don’t claim it’s perfect (sometimes it can be noisy and confuse the model), more often than not, it can provide an insight into where the SAE neuron is looking. As suggested, to qualitatively give an idea of how these heatmaps look, we added Appendix I, where we show examples of highly activating images with the heatmap overlayed exactly as it was shown to the LLM (with all concepts from Fig.3). In general, the two main interpretability tools we have for visualizing a neuron's behavior are (i) top activating images and (ii) activation maps (apart from activation maximization techniques, which we don’t consider, as it is not clear how an LLM would interpret them). So we decided to use both approaches combined, as using only (i) may not give the complete picture. For instance, it can help the model understand when to focus on the nodule/lesion, like in Fig.15s, or on what’s around it, like in concept Fig. 15l.
>
> Then your next question was “But does the LLM even care about the highlighted regions?”. Yes, it does more often than not. For instance, in this recent paper [1], an LLM Agent is used to reverse engineer neuron behavior. They experiment with synthetic neurons where heatmaps are artificially generated to check whether the LLM pays attention to them. They show that labels provided by the LLM are comparable to expert ones. As we acknowledge in our paper, we took some inspiration from this work for the concept naming step.
> Regarding the annotation step, as you mention, one could simply use the SAE layer as a bottleneck layer (like in DN-CBM), but this exposes us to problems such as stubborn polysemanticity and the fact that we can’t blindly trust neuron names because at that point we would just be doing post-hoc mechinterp.
>
> Regarding possible leakage/misalignment from the top 25 images, this is a fair point. Therefore, we decided to run an experiment (Appendix G.3) to quantify this effect if it exists. We poisoned the reference set on purpose with some images containing a different concept and checked if that concept was annotated more often than a random one. More details are in the paper, but in short, the effect is not statistically significant over 293 LLM calls. Cohen’s h effect size is way below what it’s normally considered to be small. Still, we acknowledge that absence of evidence is not evidence of absence, meaning that no single experiment can completely rule out the presence of this effect, but our experiment suggests that even if it exists, it’s tiny.
> We also want to emphasize that the current limitations of state-of-the-art CBMs are not as subtle. VLG-CBM relies on GroundingDINO for annotation, which more often than not localizes concepts even when they are not visually present based on spurious correlations. It can be verified in the demo (https://huggingface.co/spaces/merve/Grounding_DINO_demo) using an image of a stereo or speakers without volume knobs or anything resembling an electronic device, and “volume knobs” as a prompt will almost always be annotated as present. For the others that use CLIP, it may be even worse, as due to how this model was trained, we expect an image of a stereo to have high similarity to the prompt “volume knob”, independently of whether the knob is present in the image or not.
>
> Regarding the prompt for dataset annotation, all prompts used are in the anonymous repository. We also paste it here:
>
> “We're studying neurons in a neural network. Each neuron looks for a particular concept in an image. Below are 25 reference images (not to label) that strongly contain the concept '{concept_name}'. Carefully review these reference examples before starting the main task.
>
> -> 5x5 reference grid
>
> Now, here are {images_per_batch} new images (in a {grid_size}x{grid_size} grid, read left to right, top to bottom). For each image, predict '1' if the concept '{concept_name}' is present, or '0' if absent. Return your answers as a Python list of 0s and 1s in order. Only output the list."
>
> -> 5x5 image grid
>
> Regarding the top25 images grid, it can be removed, but we expect a reduction in the annotation quality, especially for less specific concepts like “long tail”. In Appendix G.1, we also added an experiment where we annotate ground truth CUB concepts with and without the top25. The difference in annotation accuracy is around 5% in favor of using the top25.

---

> ### Author Response · Authors · 2025-11-21
> **Response to Reviewer GTFX (Part 2)**
>
> W2. The point of NEC/NCC metrics and what we mean when we say that they can help in controlling information leakage can be summarized as follows: We know from previous work that less sparsity generally means more accuracy and more leakage, while more sparsity means less accuracy, less leakage (because the model has less information to reconstruct the original backbone activations), and more interpretability. With this in mind, it does not make sense to compare CBMs with accuracy alone without also considering sparsity. Therefore, it’s fairer to compare accuracies if we are at matched sparsity (i.e., same NCC or same NEC). This, however, stops working the moment you introduce leakage in the annotation step, such as VLG-CBM, therefore, it’s important to use both NEC/NCC and the random words experiments. If you combine the NCC + random words experiment, then it’s possible to see that, for instance, in Figure 2c, the random CBM has similar accuracy with the non-random ones at high NCC, and therefore, we are in a leakage scenario. However, if we select a lower NCC, then the difference between the random and non-random increases, suggesting lower leakage (but naturally also accuracy decreases a bit). Therefore, for controlling leakage, we use NCC in combination with the random words CBM. This approach should easily spot the leakage in your Identity Mapping CBM. In the vanilla CBMs where concept ground truths are known, several methods have been proposed to measure leakage (for instance, by comparing the predictive power of concept representations with their ground truth labels), but we still lack a way to apply them where such ground truth is unknown. This means that we are not able to quantify leakage in an absolute sense and neither can we completely remove it (this applies also to all baselines). To clarify this, now we mention it explicitly in the limitations.
>
> Other questions:
>
> Regarding using 5x5grid vs 1 image at a time, naturally, one at a time is better, and we also added an experiment where we test it in Appendix G.1. Using 1 image at a time obtains 5% more accuracy, but with today’s technology, this has little practical implications as doing it like this would mean 25x times more time and cost. For imagenet-scale, it would mean going from a few days to a few months for dataset annotation.
>
> Regarding elasticnet instead of lasso, we implement the sparse final layer using GLM-SAGA [2], which uses elasticnet. They explain their rationale behind using it instead of just lasso in Appendix A.2 (In short, they claim it better handles correlation between features). Since LF-CBM and VLG-CBM also use this, we decided to stick to the same training procedure for the final layer for a fairer comparison.
>
> Regarding the concepts “tiger stripes” and “yellow banana with brown spots”, these are not the only concepts that the model uses to recognize the classes. Tiger stripes are also used to a lesser extent for tiger cats, and there is also a concept that recognizes the tiger face, and bananas also have a concept to recognize them when they are green. We added these two concepts in Figure 16. Still, Imagenet classes can be pretty basic, so it may happen sometimes that one class is recognized by one or two main concepts + a few additional contextual cues.
>
> On the novelty/originality point, we acknowledge that judgments of novelty are partly subjective, but we believe our work represents a meaningful step in ante-hoc interpretability, as it provides a pipeline that turns an arbitrary backbone into a CBM that automatically learns which concepts are needed for the task. State-of-the-art without our work would consist of DN-CBM, for which it’s unclear how it could work for real-world datasets like ISIC or CUB (see its explanations in the new Appendix J), as it requires CLIP concepts (same for LF-CBM), and VLG-CBM. On this specifically, we believe our analysis on class-conditioned leakage fills an important gap in the current literature because VLG-CBM is currently used and cited as a state-of-the-art CBM, and without awareness of this limitation, practitioners may unintentionally over-trust its explanations.
> To our knowledge, we are also the first to combine Mechinterp+MLLMs for learning CBMs, and given the significant performance improvement of our M-CBM compared to baselines, it may be a promising combination for building interpretable models in the future.
>
> We believe we have addressed all of your points, but if there is still anything unclear or unconvincing, feel free to ask further questions and thank you again for the constructive review.
>
> [1] A Multimodal Automated Interpretability Agent, ICML 2024
>
> [2] Leveraging Sparse Linear Layers for Debuggable Deep Networks, ICML 2021

---

### Official Review · Reviewer_pxTa · 2025-10-30

**Soundness:** 3
**Presentation:** 4
**Contribution:** 4
**Rating:** 8
**Confidence:** 3

**Summary:**

The authors propose a new method for extracting concepts in concept bottleneck models by leveraging sparse autoencoders (SAEs) on top of a black-box architecture. They first train an SAE and identify concepts by examining the non-dead neurons. For each neuron, they select the most and least activating images from the black-box model’s dataset and use a multimodal large language model (MLLM) to assign concept labels. To train the concept bottleneck layer, they construct a semi-automated dataset containing 1,000 images per concept to determine concept presence. In addition, they introduce a softer measure of concept sparsity (NCC), with the aim of enabling more controlled and flexible comparisons across concept bottlenecks.

**Strengths:**

- The methodology is intuitive, and its benefits are evident: leveraging concepts already present in the black-box model enhances interpretability.
- The approach achieves performance improvements over baselines when controlling with NCC.
- The paper is clearly written, and most aspects of the methodology are well justified (with the exception of the weaknesses noted below).

**Weaknesses:**

While the motivation for NCC is intuitively explained (lines 303–305), it would be valuable to provide evidence for why the additional concepts are necessary. When concepts contribute substantially less than the dominant ones that NEC would capture, are they still useful and interpretable, or do they risk overfitting? NCC appears to strike a balance between interpretability and overfitting, but it is unclear how much the method benefits from this added flexibility. Would the same performance patterns hold under the stricter NEC criterion? Alternatively, is the improvement primarily due to the inclusion of negative concepts? Reporting NEC scores alongside NCC results could help clarify the extent to which this flexibility contributes to the method’s effectiveness.

**Questions:**

- While this is a clearly well-written and valuable paper, I remain uncertain about the role of NCC. One way to address this would be to also report performance under NEC; another would be to demonstrate that the extracted concepts have an intuitive and interpretable basis.
- Clarification: Beyond instructing the MLLM not to use class names as concepts, did you perform any filtering to ensure that class names were indeed excluded? (line 210)

---

> ### Author Response · Authors · 2025-11-21
> **Response to Reviewer pxTa**
>
> Thank you for taking the time to review our paper.
>
> Q1-W1. In the revised version of the paper, we give more attention to the role of NCC and its relationship with NEC. Specifically, we added Appendix H, where we go more in-depth on how NCC generalizes NEC and formally show in which circumstances they are the same. As suggested, we also provide the accuracy under NEC (which is generally about 1% lower)  and a visualization of the additional concepts that the model uses when controlling for NCC=5 instead of NEC=5 to qualitatively show that these extra concepts are also interpretable.
>
> Q2. Yes, we checked that all concept names are not exactly equal to class names. While it actually never happened that GPT4.1 used the class name as concept if asked not to do it, in principle, it may happen, especially if one uses a smaller, less performant MLLM. However, in that case, we would not necessarily suggest filtering out that concept right away, but rather try at least once to re-run the MLLM. We added more clarity on this in the revised version.
>
> We believe we have addressed all of your points, but if anything remains unclear or unconvincing, we would be happy to clarify. Thank you again for the constructive review.

---

### Official Review · Reviewer_kxfy · 2025-10-31

**Soundness:** 3
**Presentation:** 3
**Contribution:** 3
**Rating:** 6
**Confidence:** 4

**Summary:**

This paper presents a mechanistic Concept Bottleneck Mode --- M-CBM --- that brings a new opportunity to extract concepts on the fly using Sparse Autoencoders. For annotating and giving the concepts a human-readable “entity," the authors prompt multimodal LLMs, with a special accounting for compute and costs. Notably, the authors introduce a new metric for measuring the sparsiness and consequently, expressiveness of the activated concepts. This metric is based on the prior work and evaluated throughout the paper. Finally, by rigorous comparisons with popular baselines in the field, M-CBM emerges as a robust and strong baseline, facilitating future research.

**Strengths:**

**The baselines are recognized in the field and strong-performing.** The authors conduct their comparison in a simple and rigorous manner evaluating the same backbone encoder for each model on the same dataset. Considering that some datasets require a broader concept set to perform well on, the authors employ larger backbone models for all baselines, e.g., on ImageNet.

**The idea is sound and well-presented.** I like the way how authors described their framework in Sec. 3. Effort in proposing a new metric for measuring sparsity leveraging concept contribution.
Also interesting how the authors ablate their meric vs. accuracy in Fig. 2 using the baseline VLG-CBM model

**Accounting for compute and costs.** When using (M)LLMs to assign each SAE neuron to concept name or to annotate some subset of data to create a matrix of "Present" / "Missed" concept per image, the authors mention a precise cost of the OpenAI API per one annotation.
As well as stating that money spent scale linearly with the number of concepts. Thus, one can do the math and calculate the total spendings.

**Demonstrating the concept "yes / no" selection and concepts pertaining to image.** In Fig. 3, the authors show a precise example of which concepts are being selected to a particular class and which receive the negative weight. The provided concepts look very meaningful and not monosyllabic, unlike concepts in many prior works that were obtained with less sophisticated methodology.

**Weaknesses:**

**Your NCC metric.** I do appreciate the introduced metric as the extension of the NEC metric from the prior work on VLG-CBM. As far as I understand, the parameter $\tau$ controls, let us say, the "sparsity" of the impactful concepts. Meaning that the model with smaller NCC (and also the smaller NEC metric) provides more explainable decision since only few the main important concepts are activated. So it is naturally good to keep both metrics relatively small, or contrary, a large NEC (NCC) can result in information leakage. Given that both metrics are pretty similar in their essence, I would like to see more comparisons and discussion between two of them, in flavour of your Fig. 2 (or the Fig. 3 from the VLG-CBM work). Importantly, results of VLG-CBM demonstrate only minor, even negligible, drop in accuracy when their NEC is small. Which leads me to questions:
1) Does the same holds with VLG-CBM under your NCC metric?
2) Does the accuracy drop of your M-CBM is also negligible under small NCC and NEC metrics?
In my opinion, offering discussion and, which is better, experiments on this merit in your manuscript will significantly increase the reader's confidence in your NCC metric.
Also, it would be nice to include results on NCC while sweeping $\tau$. Could it be that with $\tau = 0.5$ or less many concepts will satisfy this fraction of information and, thus, the decision-making process would be rubbish? How does this connect to the fact that even with randomly initialised concept bottleneck layer the model can give near-optimal accuracy when the VLG-CBM NEC metric is large?

**Regarding your comparison in Fig. 2 (a)**, I am also curious in the following: the VLG-CBM baseline uses up to 128 concepts for CUB --- see their Fig. 3 (c). Additionally, prior work Res-CBM [1] claims that the number of concepts should not be the larger the better, meaning there is some critical k, beyond which, the accuracy will drop. Do you think this will be observed when NCC (or similarly, NEC) are very large? Offer some discussion. Ideally, it would be nice to provide figures based on your Fig. 2 with very large NCC --- at least a couple of hundreds.

**Backbone model choice.** From "Sec. 5 Setup", I see that you use ResNet-18 for CUB, ResNet-50 for ISIC1028, and ResNet-50 for ImageNet. I think that using the same pretrained backbones architectures for all baselines on the same dataset is a consistent choice and it deserves a credit for being applied --- as long as the number of total parameters (both pretrained and trainable), including your SAE is mentioned --- I would like to see a simple table summarizing this. More importantly, in Lines 369-370, you mention that DN-CBM supports only CLIP backbone and, thus, you use their CLIP ResNet-50. However, CLIP ViT-B/16 is also supported by their framework and offer better results.
Moreover, such a backbone is the widely supported option, used almost in every work on CBM. To make the results more appealing and aligned with prior literature it is better to present a version of your Table 1 with this backbone.

**Missed concepts for ISIC2018.** I am not sure wether it is my issues or yours, and apologize for potential mistake, but I cannot verify the number of concepts (and those concepts itself) for the ISIC2018 dataset, because the file `isic_concepts.txt`, unfortunately, is not found in the repo.
However, I can confirm that, mentioned in Line 377, 278 concepts for CUB, and 2648 for ImageNet are matched.

**Table 1 with the main results.** I do acknowledge that you now have different backbones, but the different in accuracy of the baselines evaluated by you in your framework with the backbones you are using differs quite dramatically from the results reported in the original papers of your baselines and their concurrent works. To give a better understanding of such a difference, I will write a precise metrics form your baselines, and their competitors that you cite but not evaluate.
For example, LF-CBM, in their original work, results in 74.31% on CUB and 71.95% on ImageNet, while the competing work --- DN-CBM --- reports for them 67.5% on ImageNet with CLIP ResNet-50 backbone and 75.4% on ImageNet with CLIP ViT-B/16 backbone. At the same time, in their paper, DN-CBM shows 72.9% on ImageNet with CLIP ResNet-50, and 79.5% with CLIP ViT-B/16 --- both results are superior to those you have mentioned for DN-CBM on the ImageNet dataset (46.71% when NCC=5, and 57.24% with NCC=avg).
Furthermore, the work you cite, but do not compare with --- Post-hoc CBM and also their model PCBM-h --- report an accuracy of 73.6% and 80.01% on ISIC, resp., which makes them better than M-CBM --- so offer a comparison or discussion regarding this.

**Comparison in concept extraction with other baselines.** It would straighten your work if you presented an analogue of Fig. 3 for several other baselines, e.g., like Fig. 3 in the very recent preprint [2].

**Missing baselines and discussions.**  Despite evaluation very popular baselines, I think you are missing comparisons and (or at least) discussions regarding the following works [1,3,4,5,6]. All of them claim to outperform the long-standing LF-CBM to some extend, and are a decent option for comparison in the field. Summarizing a ranking those in your Table with results will further guide the community towards using better and more promising baselines.

**A minor comment.** As I see, in your codebase you implement the basic Adam optimizer with the old "coupled" weight decay. I would suggest to stick to the decoupled weight decay of ~0.1 and AdamW instead of Adam.

[1] "Incremental Residual Concept Bottleneck Models", 2024

[2] "Graph Integrated Multimodal Concept Bottleneck Model", 2025

[3] "Language in a Bottle: Language Model Guided Concept Bottlenecks for Interpretable Image Classification", 2023

[4] "Sparse Concept Bottleneck Models: Gumbel Tricks in Contrastive Learning", 2024

[5] "The Decoupling Concept Bottleneck Model", 2024

[6] "Improving Concept Alignment in Vision-Language Concept Bottleneck Models", 2024

**Questions:**

See the **weaknesses** part

**Details Of Ethics Concerns:**

I do not have any major concerns. But would highlight only one, regarding Fig. 9 from Appendix, which contains some sensitive pictures of different diseases from the medical dataset. However, those are minor in my opinion.

---

> ### Author Response · Authors · 2025-11-21
> **Response to Reviewer kxfy (Part 1)**
>
> Thank you for taking the time to review our paper.
>
> W1. On the fact that VLG-CBM’s accuracy is insensitive to NEC/NCC, it is not something good, but rather a sign of leakage at the annotation level. Intuitively, if a CBM performs almost the same as the black-box using only 1 concept per prediction, it means that there is at least one concept per class that is exactly the same as the class. This happens in VLG-CBM because annotation is class-conditioned (it associates concepts to classes and only annotates the concept as present on images of that class). In Fig. 2 and Fig.7, we show indeed that with this type of annotation, even using random words as concepts yields the same accuracy.
> In healthy conditions, we expect a trade-off between accuracy and NEC/NCC (see Fig. 2b, where we remove class conditioning from VLG-CBM and Fig. 2c with our M-CBM).
> We used the “random words as concept” approach because comparing with just a randomly initialized CBL would not expose leakage if it happens at the annotation level.
> Furthermore, in the revised manuscript, we added Appendix H, where we go more in-depth on how NCC generalizes NEC and formally show in which circumstances (at which tau) they are the same. We also provide the accuracy under NEC (which is generally around 1% lower) and a visualization of the additional concepts that the model uses when controlling for NCC=5 instead of NEC=5 to qualitatively show that these extra concepts are also interpretable. Changing tau would have the same effect as increasing NCC, because we would select a less sparse model. Setting a tau=0.5 is extremely low, as it could be interpreted as “on average, the top contributing concepts explain half of the prediction”. We can think of NCC and tau as two knobs that we can tune to control the interpretability-accuracy trade-off. In the paper, we fix tau=0,95 and only tune NCC.
>
> W2. The total number of concepts (i.e., the dimension of the bottleneck) in the CBL, which we refer to as K, is the maximum possible value of NCC and NEC. In Fig.3c of the VLG-CBM paper, 128 is NEC (which is the average number of concepts per class that have non-zero weights in the final classifier), which is different from the CBL size. We were not able to find in the Res-CBM paper specifically this statement that increasing K can make accuracy drop at some point. As we understood it, in the “Rethinking CBMs” paragraph, they say that if K is greater than or equal to the backbone dimension d, then increasing it further would reduce efficiency and interpretability without increasing accuracy. This aligns with results from prior works, and we discuss this at the beginning of Section 4 (their d is our n). We added Res-CBM as an additional reference there, as it provides further evidence. This problem, where, with a sufficiently high K, the classifier can easily re-estimate the backbone activations and cause leakage, only applies with a dense classifier, and this is where metrics that measure sparsity like NEC/NCC come in. This is also well explained in the VLG-CBM paper with their analysis on the randomly initialized CBL. Coming back to NCC and Fig. 2, since NCC measures how many concepts on average are used per prediction, by definition, it cannot be higher than K. For CUB, our K is 278, so that Figure could potentially extend up to 278, but accuracy saturates much earlier. Therefore, we show up to 100 and made the scale logarithmic to better show the behaviour at low NCC, which we are mostly interested in. Regarding the possibility of the accuracy dropping with higher NCC, we have never experienced this and nor do we expect it to happen, since allowing more active concepts simply gives the classifier more degrees of freedom, so accuracy typically saturates rather than drops. What degrades as NCC grows is efficiency, interpretability, and information leakage.
>
> W3. For all baselines (except DN-CBM), we use the same exact backbone, not only with the same amount of parameters but also with the same weights. The SAE is used only for concept extraction, and it’s not part of the final M-CBM model, so it does not affect the parameter count. The only thing that affects the number of parameters other than the backbone is the size of the CBL (K). In Appendix, we added Table 5 as suggested with the total number of parameters for each model used in the paper. Regarding DN-CBM backbone, since it requires a CLIP backbone, we initially used ResNet50 for consistency with the other baselines. Now, in the revised version, we also added CLIP ViT-B/16 as suggested. The results are better but still below the other baselines. Still, we want to note that if we had used a ViT backbone for M-CBM (or any other baseline), our results may have been better too. However, we mainly wanted to stick to the same pre-trained backbones used in prior works.
>
> W4. We checked the repository, and we are able to see this file under the following path: M-CBM/concept_files/isic_concepts.txt.

---

> ### Author Response · Authors · 2025-11-21
> **Response to Reviewer kxfy (Part 2)**
>
> W5. As mentioned above, we do use the same backbones as prior work. The key difference is that Table 1 always compares models at matched sparsity via NCC because accuracy is strongly tied to sparsity. Dense classifiers (high NCC/NEC) are more accurate but leak more information and are less interpretable, while sparse ones (low NCC/NEC) are less accurate but more interpretable. Thus, it is not meaningful to compare accuracies without considering sparsity.
> The fair comparison is at the same NCC/NEC. For example, LF-CBM reports 71.95% ImageNet accuracy in a very low-sparsity regime, but VLG-CBM re-evaluate LF-CBM at NEC=5 and obtains 60.30%. Under the same backbone, concepts, and training, we obtain 62.20% at NCC=5. The only difference is the enforced sparsity in the last layer. For instance, at NCC=30 (less sparsity so less interpretability), LF-CBM achieves 67.31% accuracy on ImageNet, while M-CBM achieves 74.50% accuracy. Figures 16/17/18 in the LF-CBM paper can give an intuition of why low sparsity is so problematic: when the contribution of the “Sum of all other features” is substantial then the model is difficult to interpret and ideally we want that to be very close to 0. Their Figure 8 is also another extreme example where sparsity is completely removed.
> Regarding Post-hoc CBM and PCBM-h, their ISIC numbers correspond to a binary malign vs benign setting, while we follow the standard 7-class ISIC2018, so accuracy are not comparable. Furthermore, Post-hoc CBM uses manually defined concepts and curated example images, and PCBM-h further adds a non-interpretable component to boost accuracy. There is a version of PCBM that does not require manual concepts annotations using CLIP, but this version was outperformed already by LF-CBM and it’s also not scalable to larger datasets.
>
> W6. We agree. We added 3 figures (one per dataset) in Appendix J similar to the one in the mentioned preprint. It should add value by qualitatively showing the differences between the types of concepts used by each method.
>
> W7. As we understood, [6] assumes access to expert labels and focuses on aligning CLIP concept scores with them, while [4] resembles LF-CBM but enforces sparsity at the CBL level rather than in the final classifier. For these two papers, however, we could not confirm they are published/peer reviewed. [5] shows better performance than LF-CBM in their VLM-based version but it mainly comes from adding an uninterpretable component similar to PCBM-h. Furthermore, their official codebase says “The vision-language-model (VLM) part is being refined and will be available soon”. ResCBM [1] also requires CLIP and shows only modest gains (~1–2%) over LF-CBM on CIFAR and reports 62% accuracy on CUB but unclear scalability to more complex datasets like ISIC or ImageNet. Furthermore, they also use non-interpretable components. LaBo [3] similarly relies on CLIP and in the VLG-CBM paper it performed very poorly at NEC=5 (24% accuracy on ImageNet and 41% on CUB), significantly below what we achieve under NEC=5 (see Table 7). We can include it in our Table but do not expect it to be particularly competitive in more demanding scenarios like ISIC. Furthermore, LaBo was already in our related works and now we also added [1] and [5].
>
> W8. Yes, we used the basic Adam, as this was also the default of the other baselines and we tried to keep hyperparameters the same for fair comparison. Based on your suggestion, we tested AdamW on CUB and accuracy was basically the same. Still, in the final codebase, we will definitely include AdamW among the selectable optimizers.
>
> We believe we have addressed all of your points, but if there is still anything unclear or unconvincing, feel free to ask further questions and thank you again for the constructive review.

---

### Official Review · Reviewer_gj53 · 2025-10-31

**Soundness:** 2
**Presentation:** 2
**Contribution:** 2
**Rating:** 4
**Confidence:** 4

**Summary:**

The paper proposes M-CBM, a pipeline that (i) trains an SAE on a frozen backbone’s features, (ii) uses an MLLM to name SAE neurons, (iii) asks the MLLM to annotate concept presence/absence on a subset of images (partly from test), and (iv) trains a CBL and a sparse linear classifier. The authors also introduce NCC (Number of Contributing Concepts) as a decision-level sparsity metric.

**Strengths:**

1. Clear, modular pipeline with concrete engineering choices. Training of the CBMs is standard and well-described.
2. NCC formalizes decision-level sparsity (concept contribution × weight) and can be computed per image/class.
3. Results tables/curves are easy to read.

**Weaknesses:**

1. Methodological leakage / evaluation contamination.
The paper explicitly annotates concepts on 20–30% of the test set (per concept) “solely for a final CBL evaluation.” But concept labels on test are produced using the same pipeline (SAE activation pre-selection + MLLM conditioned on highly activating images). This risks circularity and optimistic estimates of “concept learnability/consistency,” since test annotations are themselves guided by backbone activations and neuron saliency that were derived from training. At minimum, this violates a clean separation of test supervision and model analysis and undermines the claimed generalization of the concept predictor.
2. Reproducibility concerns due to proprietary dependencies.
Core steps depend on GPT-4.1 for naming/annotation and OpenAI embeddings for merging, with specific costs/timings reported; there is no open substitute evaluated. Reproducibility and fairness of comparison suffer because model behavior can change with closed APIs and prompts, and the paper does not provide robustness to alternative open MLLMs/embeddings.
3. Novelty over prior work is incremental.
DN-CBM already uses SAE-based concept discovery and naming; M-CBM mainly claims backbone-agnosticism and a different annotation path. NCC, while useful, is a straightforward generalization of NEC from weight-count to decision-level contribution.
4. Baselines and fairness.
The paper drops the original (class-conditioned) VLG-CBM from main comparisons on grounds of leakage, and reports VLG-CBM{CA} as the “fair” variant—yet **ImageNet-scale VLG-CBM{CA}** is marked N/A, while M-CBM still uses thousands of GPT annotations, a heavy but different cost. The choice of backbones is also uneven (DN-CBM forced onto CLIP-RN50 across all datasets), which can disadvantage it (reported accuracy much lower than original paper) and muddle conclusions about accuracy–sparsity trade-offs.
otential confirmation bias in concept discovery.
5. Concept naming/annotation is conditioned on highly activating examples of the SAE neuron and saliency computed from SAE decoder weights. This can bias the MLLM toward naming whatever the neuron already encodes, rather than establishing human-grounded semantics independent of the backbone.
6. Inconsistent focus on “concept leakage” without any quantitative metric.
The authors repeatedly stress concept leakage but never evaluate it quantitatively. The proposed NCC only measures sparsity, not leakage strength or label information flow, so their claims of “leakage control” remain unsubstantiated.

**Questions:**

1. How do you prevent test-time contamination when concept labels on test are produced via the same activation-conditioned pipeline (SAE saliency + top-activating grids)? Please clarify precisely which model components see any information derived from test images and at what stage.
2. Can you re-run the full pipeline without proprietary models (e.g., open-source MLLMs/embeddings) and report gaps in naming/annotation quality, costs, and final NCC-matched accuracy?
3. Provide sensitivity of your results to (a) SAE expansion factor, (b) pruning tolerance, (c) activating vs non-activating examples per neuron during naming/annotation.
4. For fairness, could you supply ImageNet-scale VLG-CBM_{CA} with cost-matched budgets (e.g., same wall-clock as your GPT calls), or conversely report M-CBM results with the same annotation budget as VLG-CBM_{CA}?
5. On the absence of an explicit concept-leakage metric (e.g., Impurity Score).
The manuscript highlights concept leakage as a major challenge motivating M-CBM, yet the evaluation currently focuses on sparsity (NCC) rather than a dedicated leakage measure. It might strengthen the empirical section to include or discuss a quantitative leakage metric, for example, the Impurity Score or a mutual-information-based variant. Such an analysis would help clarify whether the observed improvements stem from genuine leakage mitigation or primarily from sparsity regularization.
6. On comparison with CBMs that explicitly address concept leakage.
Several works have proposed concrete strategies to quantify or reduce concept leakage. Positioning M-CBM relative to these methods, either through direct comparison or a discussion of conceptual differences, could provide valuable context and highlight the distinct contributions of your work beyond sparsity control.

---

> ### Author Response · Authors · 2025-11-21
> **Response to Reviewer gj53 (Part 1)**
>
> Thank you for taking the time to review our paper.
>
> W1–Q1. Concept labels on the test set are never used to train any part of M-CBM. The backbone, SAE, concept predictor, and final classifier are all trained using only training data. Test concept labels are generated afterward and used solely in the “Evaluating Concept Prediction” section. You are correct that the numbers in Table 2 are optimistic in an absolute sense, because both train and test labels come from the same annotation pipeline. As we note in the paper, these metrics measure consistency of the annotator and learnability rather than agreement with human ground-truth concepts. However, the comparison with VLG-CBM is still fair, because its CBL is evaluated using exactly the same approach, i.e., concept labels on train and test are produced by the same annotation pipeline (GroundingDINO).
>
> W2-Q2. Yes, we tried to re-run the pipeline again on CUB (which is the smallest dataset) using InternVL3.5-241B-A28B (and e5-large-v2 for the merging step) as MLLM, which is a recent open source model that supports multi-images inputs. We obtained an accuracy of 71.40% at NCC=5 and 74.02% at NCC=avg. The ROC-AUC of the concept predictions was 74.49% and 58.14% for the worst 10%. As also discussed above, these results are obtained by annotating the test set with the same pipeline and we use them as a proxy of annotation quality since we do not have a ground truth. Also the naming was slightly worse because concepts were more generic and therefore more were merged so at the end we had 195 instead of the 278 we had with GPT4.1. The conclusion is that the annotation quality is lower and it's somewhat halfway between M-CBM with GPT4.1 and VLG-CBM_CA with GroundingDINO. Same for the final accuracy, which is lower than M-CBM with GPT4.1 but still higher than all other baselines. Qualitatively we see clearly that the annotation quality it's worse, but GPT4.1 was not perfect either. This suggests that as expected the performance depends on annotation quality, but this also means there is large space for improvement (GPT4.1 is not state-of-the-art anymore and there are many more powerful models available such as GPT5.1 or Gemini3). We will run the pipeline again using InternVL3.5 also for ISIC2018 and ImageNet and add an Appendix section with the results in the camera-ready version (in case the paper is accepted).
> Regarding the reproducibility of the GPT4.1 runs, we will release the full set of annotations.
> Lastly, for ISIC we tried to name a few neurons using MedGemma, but it consistently fails as it is not able to deal with more than one image.
>
> Q3. We agree that these hyperparameters are important, but a sensitivity study in which we repeat the entire pipeline end-to-end for different SAE expansion factors, pruning tolerances, and annotation settings would be very expensive and time-consuming in our setup. Annotating ImageNet alone already took several days of continuous computation, and repeating this multiple times for different hyperparameter grids could take months, ignoring monetary cost. Still, we’ll try to briefly explain the rationale behind our SAE hyperparameters (also detailed in Appendix B). For the expansion factor, we cannot practically go beyond 4x at ImageNet scale, as this would make the number of concepts (and thus annotations) computationally infeasible with current models. On CUB and ISIC, we did try higher expansion factors and found that only the number of dead neurons increases meaningfully. Intuitively, while ISIC and ImageNet share the same backbone size, the number of discoverable concepts is not the same. For ISIC we could not get more than about 150 non-dead neurons even when increasing the expansion factor, so using the same expansion factor as ImageNet would not be meaningful.
> For pruning, we were very conservative. Besides dead neurons, Fig. 5 shows that most of the pruned neurons have extremely low frequency (on the log scale, −3 means they are active in about 0.1% of the training set, so literally 10 images for ISIC or 5 for CUB). Keeping such rare concepts could make multi-label training unstable. The pruned neurons do not contribute significantly to performance either as when we remove them, loss and accuracy barely move and sometimes even improve slightly (see Appendix B). Qualitatively, we found that these pruned neurons also tend to look like noise.
> Regarding the use of highly activating images during annotation, we ran an experiment (Appendix G.1) on CUB ground-truth attributes where we apply our annotation pipeline with and without a reference grid, and we observe about a +5% improvement in annotation quality when the reference is included. This suggests that providing examples may be beneficial, especially for concepts like “long tail” or “small nevus”, where it could be useful to provide examples to calibrate what “long” or “small” mean in the context of that backbone.

---

> ### Author Response · Authors · 2025-11-21
> **Response to Reviewer gj53 (Part 2)**
>
> W3. On the novelty point, we acknowledge that judgments of novelty are partly subjective, but we believe our work represents a meaningful step in ante-hoc interpretability, as it provides an automated pipeline that turns an arbitrary backbone into a CBM without having to know a-priori the concepts that are needed for the task. State-of-the-art without our work would consist of DN-CBM, for which it’s unclear how it could work for real-world datasets like ISIC, as it requires CLIP concepts (same for LF-CBM), and VLG-CBM. On this specifically, we believe our analysis on class-conditioned leakage fills an important gap in the current literature because VLG-CBM is currently used and cited as a state-of-the-art CBM, and without awareness of this limitation, practitioners may unintentionally over-trust its explanations.
> To our knowledge, we are also the first to combine Mechanistic Interpretability+MLLMs for learning CBMs end-to-end that automatically discover the concepts needed for the task, and given the significant performance improvement of our M-CBM compared to baselines, it may be a promising combination for building interpretable models in the future.
> Furthermore, this may also be a subjective statement, but by looking at Table 1 and the improvements we obtain over the state-of-the-art especially at NCC=5 (which is what really matters for interpretability), we view these gains as more than incremental. We also added Appendix J to show qualitatively the difference between our concepts and DN-CBM.
>
> W4-Q4. Comparing with the original VLG-CBM would not be informative as it can achieve close to black-box accuracy regardless of the concepts used and the sparsity level. As we show in Fig. 2 (and Fig. 7 in Appendix B), even with random concepts it predicts as well as the backbone. We show in the paper that this comes from the class-conditioned annotation which causes severe leakage. For each class, a set of concepts is selected and annotated only on images of that class, and the concept is set as “absent” elsewhere. Together with the high false positive rate of GroundingDINO (see Appendix D in VLG-CBM paper), this makes even random concepts behave as class proxies. Now, instead of dropping this approach completely, we tried to fix the leakage issue by proposing the VLG-CBM_{CA} variant which is simply VLG-CBM without class conditioning. Instead of assigning concepts to classes, we annotate all images in a class-agnostic way. This removes the class-conditioning leakage as we show in Fig. 2 where the random VLG-CBM_{CA} now behaves similarly to the random M-CBM, which is also class agnostic. However, this makes the cost of VLG-CBM_{CA} enormous at ImageNet scale as you have millions of images and thousands of concepts. Using an H200 (which is respectable hardware), we found that in the same amount of run-time that we used to annotate ImageNet for M-CBM, we would have annotated only about 1% of ImageNet with GroundingDINO, i.e., roughly 10k images (10 per class). While M-CBM can naturally work with a subset because the SAE suggests which images to annotate for each concept, it is unclear how to extend VLG-CBM_{CA} to also work with such a small subset because the selection would essentially have to be random. We therefore do not expect such a severely under-annotated setting to give a meaningful evaluation of VLG-CBM_{CA}. Furthermore, regarding ISIC and CUB, VLG-CBM_{CA} required more time than M-CBM. Regarding the monetary cost, running locally on a rented H200 typically would cost about the same as using GPT4.1 API per hour as each concept took on average 2minutes and USD 0.14, so roughly USD 4/hour.
> On DN-CBM, using CLIP is not our choice but a requirement of the method. We now have added results of DN-CBM also using a ViT-B/16 backbone which improves accuracy especially on ImageNet, but relatively to the other baselines it remains the weakest therefore the conclusions do not change. The higher accuracy reported in the DN-CBM paper is obtained at lower sparsity. As also shown in Appendix C of the DN-CBM paper, there is a clear sparsity-accuracy tradeoff. Therefore, in our paper, all CBMs are evaluated at the same sparsity budget (NCC=5 and the average accuracy between NCC at levels 5,10,15,20,25,30).

---

> ### Author Response · Authors · 2025-11-21
> **Response to Reviewer gj53 (Part 3)**
>
> W5. Our goal is not to define concepts independently of the backbone. By design we want concepts to be the human-understandable version of what the SAE encodes which is what the backbone learned during training. Furthermore, we want that such human-understandable concepts are as close as possible to the feature encoded in the SAE neuron because those features are both present in the data and have predictive power for the task.
> A possible concern is whether this conditioning could cause the MLLM to overfit to idiosyncrasies of the top-activating images (i.e., annotate an image as positive just because it activates the SAE neuron and is in the reference set, even though it is not aligned with the concept semantics). To investigate this, we added an experiment in Appendix G.3 where we intentionally poison the top-activating images with examples of a different concept and measure how often the MLLM annotates this spurious concept compared to what we would expect from random chance. The effect is very small and not statistically significant, suggesting that while the reference set helps the MLLM understand what the neuron encodes, it does not bias it toward annotating images that are not aligned with the concept semantics.
>
> W6-Q5-Q6. We agree that NCC/NEC are not dedicated leakage metrics. We use them to control sparsity which is known from prior work to correlate with leakage. In general, less sparsity means more accuracy and more leakage, while more sparsity means less accuracy, less leakage (because the model has less information to reconstruct backbone activations), and more interpretability. However, it does not guarantee the absence of leakage (we clarify this better now in the limitations), especially if it comes from the annotation step. For instance, the vanilla VLG-CBM suffers from high leakage at very low NCC. To control leakage, we combine NCC with the “random words as concepts” approach. If a CBM trained with random concepts obtains similar accuracy to the same model trained with discovered concepts, then the performance cannot be attributed to the concepts and we interpret this as evidence of leakage. For instance, in Fig. 2c the random CBM has very similar accuracy to the non-random one at high NCC. As we lower NCC, the gap between random and non-random increases, suggesting lower leakage. One then can choose to select a model at a NCC level where the gap is large and this is what we mean when we say “control leakage” with NCC.
> Regarding dedicated leakage metrics such as Impurity Score or mutual-information-based measures, they were developed for the classic CBM setting where concepts are part of the dataset and have ground-truth labels. The main idea behind such metrics is that you can compare the predictive power of ground truth labels versus the concept representations and if the latter are more predictive then this can be attributed to leakage.
> In our setting (and also in LF-CBM, DN-CBM, VLG-CBM), concepts do not have a ground-truth. If for instance we tried to apply such a metric to VLG-CBM using its GroundingDINO based annotations as “ground truth”, we would likely conclude that leakage is absent, because the annotations would predict the classes much better than the representations. Yet this would be misleading as our analysis shows that VLG-CBM can reach near–black-box accuracy even with random concepts and therefore leakage must be very high. As suggested, we added a discussion in related work regarding these metrics and why they are conceptually hard to apply in our setting.
>
> We believe we have addressed all of your points, but if anything remains unclear or unconvincing, we would be happy to clarify. Thank you again for the constructive review.

---

> > ### Comment · Reviewer_gj53 · 2025-11-27
> >
> > Thanks for the rebuttal, I have raised my score to weakly accept. Howerver, I still have a few questions.
> >
> > W1-Q1.  As [1] stated, there are several issuses when use VLG-CBMs. So I am curious about if M-CBM consider or can sovle these problems?  VLMs are not comparable with human experts, and VLM-CBMs tend to be less interpretable than CBMs.
> >
> > W2-Q2 suggestions. Medical concept can use other medical open-sourced LLMs (e.g., PanDerm [2], Lingshu [3], Hulu-Med [4])
> >
> > BTW, if the authors highlight their changes in the revision will be easier to compare with the previous version.
> >
> > [1] Debole, Nicola, et al. "If Concept Bottlenecks are the Question, are Foundation Models the Answer?." arXiv preprint arXiv:2504.19774 (2025).
> >
> > [2] Yan, Siyuan, et al. "A multimodal vision foundation model for clinical dermatology." Nature Medicine (2025): 1-12.
> >
> > [3] Xu, Weiwen, et al. "Lingshu: A Generalist Foundation Model for Unified Multimodal Medical Understanding and Reasoning." arXiv preprint arXiv:2506.07044 (2025)
> >
> > [4] Jiang, Songtao, et al. "Hulu-med: A transparent generalist model towards holistic medical vision-language understanding." arXiv preprint arXiv:2510.08668 (2025).

---

> > > ### Author Response · Authors · 2025-11-28
> > >
> > > Thank you for taking the time to read our rebuttal and for increasing your score.
> > >
> > > Q1. Thank you for sharing this very interesting preprint. We agree that annotations provided by today’s VLMs are generally inferior to human experts, especially if their expertise is high, and this gap can affect interpretability. However, manual annotation of hundreds or thousands of concepts over large datasets is rarely feasible and not scalable. Our M-CBM significantly improves over VLG-CBM on annotation quality, and it also also modular by design, meaning that the MLLM can be easily swapped with a newer one that may be released in the future, which would further increase annotation quality. In principle, the MLLM could even be swapped with a human expert, and the manual work would still be significantly lower than manually annotating the whole dataset. Furthermore, regarding the poor quality of the concepts used by VLG-CBM, this is also consistent with our results. Fig. 20 illustrates that VLG-CBM often uses concepts that are non-visual or absent from the dataset (e.g., cancer, sun-damaged skin, pathological entity), whereas M-CBM learns concepts directly from the data, making it more visually grounded and interpretable than prior VLM-CBMs. Furthermore, the M-CBM paradigm, where concepts are learned automatically, could even have advantages over human-defined ones, potentially allowing human experts to learn something new from ML models, especially in cases where these models may surpass human capabilities.
> > >
> > > Q2. Thank you for the suggestions. Our MLLM annotator needs to (i) take images as input, (ii) handle multiple images per prompt, (ii) generate textual concepts or yes/no labeling as output. Not all the suggested models satisfy these requirements (e.g., PanDerm is a vision backbone), and Hulu-Med (which was released just a month ago) seems to be the most promising, as on benchmarks it seems to surpass MedGemma (which we already tried) and InternVL3.5, but still it’s not superior to GPT4.1. As we also mention above, our pipeline is modular, so any MLLM that meets these requirements can be plugged in. We will definitely try to test Hulu-Med for ISIC as an open-source alternative to GPT4.1.
> > >
> > > On highlighting the changes, to the best of our knowledge the conference system applies a pdfdiff to compare new changes to the paper against the original submission, however, it seems that it is still not visible. To facilitate comparison, we will soon write a general comment with a comprehensive list of all changes made and all newly added appendix sections.

---

### Author Response · Authors · 2025-12-03
**General Comment**

We thank all the reviewers for their constructive feedback and useful suggestions. In addition to the detailed responses to each review, this general comment summarizes the additional experiments and changes we made in the revised version based on the reviewers’ feedback and questions.

We added Appendix I to show examples of top-activating images for SAEs with the heatmap overlaid exactly as it is shown to the MLLM (Reviewer *GTFX*).

We added Appendix G with three new experiments to quantify (i) the benefit of using the reference set in terms of annotation quality, (ii) the trade-off in annotation quality between annotating images one at a time or 25 at a time using the 5x5 grid, and (iii) the robustness of the annotation under reference set poisoning (Reviewer *GTFX and gj53*).

We added Appendix H, where we go more in depth on the role of NCC and formalize its relationship with NEC, showing also accuracies under NEC as well and examples of the additional concepts that the model is allowed to use when controlling for NCC=5 instead of NEC=5 (Reviewer *pxTa and kxfy*).

We added Appendix D, where we detail in a table the total number of parameters for each model and backbone used in the paper (Reviewer *kxfy*).

We added DN-CBM with a CLIP ViT-B/16 backbone in Table 1 as an additional baseline (Reviewer *kxfy and gj53*).

We added Appendix J to qualitatively show the differences between the concepts used by different methods (Reviewer *kxfy*).

We expanded the related work section to include the additional suggested references (Reviewer *kxfy*).

We added a discussion (in related works) of leakage mitigation metrics for classic CBMs, explaining that due to the lack of ground truth concepts, these metrics do not directly apply to methods like M-CBM, DN-CBM, VLG-CBM and LF-CBM (Reviewer *gj53*).

We added the option in the codebase to use both AdamW and Adam (Reviewer *kxfy*).

We re-ran the full pipeline on CUB using an open-source MLLM (InternVL3.5-241B-A28B), obtaining an accuracy of 71.40% at NCC=5 and 74.02% at NCC=avg, which are slightly worse than using GPT4.1 (as expected) but still better than all baselines. We are also currently running the same experiment for ISIC2018 and ImageNet, and will add an appendix section with the results in the camera-ready version in case the paper is accepted (Reviewer *gj53*).

---

### Meta-Review · Area_Chair_Tkcu · 2025-12-24

**Summary:**

This paper was reviewed by four experts in the field, and the reviews are mixed. The paper received one Good Paper (8) score, one Marginal Accept (6) score, and two Marginal Reject (4) scores.
The paper proposes a novel CBM technique called Mechanistic-CBM (M-CBM). The techniques main novelty lies in the fact that the technique uses a sparse autoencoder (SAE) to discover concepts from a frozen backbone. For each neuron of the SAE, the authors retrieve the images that most strongly activate it and also the image that non-activated the neuron and then the technique uses a MLLM to assign a concept name. They then use the MLLM to annotate a small subset of the dataset with binary labels indicating the presence or absence of each concept in an image. Using these concept-annotated images, the authors train a CBM. They also introduce a new metric called the Number of Contributing Concepts (NCC), presented as a more flexible alternative to NEC for measuring concept sparsity. Overall, M-CBM is presented as a method for transforming a wide range of backbone models into CBMs.
Based on the reviews, I side with the reviewers recommending acceptance.

**Reviewer Concerns:**

The reviewers raised several important concerns that can be summarized as follows:
1. Leakage: One shared concern is test-set contamination. Concepts are annotated on a small portion of the test set, and these labels are generated via the SAE pipeline, which can blur the boundary between analysis and evaluation. Reviewers also stress that NCC does not capture leakage, and they ask for additional metrics that more directly probe leakage.
2. Reproducibility: Reviewers were concerned about the reproducibility of the pipeline, as it relies on the GPT-4.1 MLLM. They request the use of open-source MLLMs.
3. Baselines and backbone choices: Reviewers flag uneven backbone choices, such as using CLIP with the ResNet-50 backbone, and note that some baseline accuracies differ from those reported in the original papers or concurrent works. They ask for either stronger, better-matched baselines or clearer explanations for these discrepancies.
4. Annotation transparency: Reviewers also ask for more transparency in the MLLM-based annotations, including example overlays shown directly, the exact MLLM prompts, and ablations testing how annotations change with and without the 5×5 grid used in the annotation process. They were also concerned that some concepts could inherit class names.
5. NCC vs. NEC: A main theme of the discussion was what NCC adds beyond NEC, and whether the low-contribution concepts admitted by NCC remain interpretable or instead introduce overfitting. Reviewers request NEC results alongside NCC results and clearer evidence that NCC is a more flexible and improved metric.

**Reviewer Scores:**

The initial ratings were Marginal Accept (6), Good paper (8), and two Marginal Reject (4). The  Reviewer gj53, who initially assigned a rating of 4, revised it to raise it to 6, as he indicated in openreview. I don't think the other reviewers would have increased their ratings, especially since the scores were already high.

After the rebuttal, the authors have, in my opinion, addressed most of the points raised. Therefore, the article should be accepted.

---

### Decision · Program_Chairs · 2026-01-26

Accept (Poster)